# High-Resolution Terrain Reconstruction of Slot Canyon Using Backpack Mobile Laser Scanning and UAV Photogrammetry

**Yonghui Xin [1], Ran Wang [1,*], Xi Wang [1], Xingwei Wang [1], Zhouxuan Xiao [2] and Jingyu Lin [3]**

1    School of Earth Science and Resources, Chang'an University, Xi'an 710054, China
2    School of Geospatial Engineering and Science, Sun Yat-sen University, Zhuhai 519082, China
3    Faculty of Geographical Science, Beijing Normal University, Beijing 100875, China
*    Correspondence: wangran@chd.edu.cn

**Abstract:** Accurate terrain models are critical for studying the formation and development of slot canyons. However, for slot canyon landforms, it is challenging to generate comprehensive and high-resolution morphological data by individual observation due to the inaccessibility of steep walls on either side and the complexity of the field observation environment, such as variable-slope terrain, partial vegetation cover, and lack of satellite signal. Off-the-shelf surveying techniques, including Unmanned Aerial Vehicles (UAV) photogrammetry and Backpack Mobile Laser Scanning (BMLS), facilitate slot canyon surveys and provide better observations. This paper proposes an integrated scheme to generate comprehensive and centimeter-resolution slot canyon terrain datasets (e.g., color point clouds, Digital Elevation Models (DEM), and 3D mesh) using BMLS and fine UAV photogrammetry. The results show that the fine flight of UAVs based on a rough model can avoid collision with obstacles or flying into restricted areas, allowing users to perform tasks faster and safer. Data integration of BMLS and UAV photogrammetry can obtain accurate terrain datasets with a Root Mean Squared Error (RMSE) of point cloud registration of 0.028 m. Such high-resolution integration terrain datasets reduce local data shadows produced solely by individual datasets, providing a starting point to revealing morphological evolution and genesis of slot canyons.

**Keywords:** slot canyons; point clouds; DEM; backpack mobile laser scanning; UAV photogrammetry; data integration; terrain

## 1. Introduction

A slot canyon is a valley landform with a high aspect ratio, manifested by the valley side being much larger than the valley bottom, and is morphologically distinct from both loess valleys and rock gorges [1–4]. They have unique morphological characteristics in three aspects [5]: (1) Narrow at the bottom and sometimes wide at the top. The wide top of the slot canyons reaches tens of meters, but the width at the bottom is nearly tens of centimeters, allowing only one person to pass. (2) Steep sidewalls. Variable slopes and a large number of high-angle planes are characteristics of slot canyons. (3) Partial vegetation cover. At the top, more quaternary deposits remain due to the gentle slope, supporting vegetation development, while at the bottom, more bedrock is exposed owing to the steep slope, leading to sparse vegetation. Generally, this type of landform occurs when valleys are connected by gorges [6], which is often represented by the valley-in-valley landform and one-sky landscapes, such as Antelope Canyon on the Colorado Plateau, USA, and the Slots in the Chinese red beds landscapes. They have significant scientific and tourism value. Additionally, slot canyons exhibit many dynamic geomorphological phenomena associated with mass movements, such as landslides and erosion, which often alter the morphology of the terrain surface [7]. Therefore, capturing the impact of these processes requires a flexible mapping workflow that results in the digital terrain dataset having sufficient spatial detail and measurement accuracy in capturing the fine-scale morphology.

In traditional studies on terrain observations, the most common methods can be divided into single-point-based models and satellite digital elevation models (DEM). In conjunction with Global Navigation Satellite Systems (GNSS), it is possible to easily position survey work within a global coordinate system by utilizing single point-based methods using level gauges, theodolites, and total stations [8]. The accuracy and precision of such methods can be down to the cm or mm scale, but they also tend to give limited, fragmented data and generate data with wildly varying spatial density and uncertainty. In addition, getting data dense enough to make terrain data would take a long time [9]. There is also much research on large-scale terrain and geomorphology analysis using DEM data from satellites [10,11]. However, their resolution from tens of meters to sub-meters can only describe large medium-sized landforms, but cannot accurately describe small micro-landscapes such as terraces, potholes, ledges, faults, and fractures [12].

Recent advances in geospatial data collection, especially the application of active laser scanning and passive photogrammetry-based methods, have significantly improved the ability for high-resolution terrain research [13–15]. Laser scanning, as a non-contact active measurement method, has the ability to sample several kinds of surfaces (e.g., top of vegetation canopy, inter-canopy surfaces, and ground) that are in the line of sight of the laser beam until impermeable surface restrains further penetration of the laser energy [16]. Photogrammetry-based methods also have been increasingly applied in the geosciences due to their low costs and high efficiencies [17–20]. In addition to obtaining point clouds with spatial coordinates (X, Y, Z), photogrammetry data also provide a large amount of texture information that can distinguish the objects. Restricted lines of sight in slot canyons severely impede the use of the widely used Terrestrial Laser Scanning (TLS) technique, and data acquisition is spatially limited due to its static nature, requiring multiple adjustments to achieve full coverage [9]. Surveys using Backpack Mobile Laser Scanning (BMLS) or Handheld Laser Scanning (HLS), typically used by sophisticated Simultaneous Localization and Mapping (SLAM) algorithms, avoid the problem of multiple adjustments of TLS. They can capture information on vertical surfaces (e.g., cliffs, steep slopes, landslides) at the canyon bottom without satellite signal environments [14,21], but rarely obtain sample boundaries across the upper zones within the field of view (e.g., rock shelters, building roofs), limiting the acquisition of complete data. Aerial laser scanning and aerial photogrammetry can obtain the top data through Unmanned Aerial Vehicles (UAV) systems. Compared to aerial laser scanning, the results of UAV photogrammetry contain more texture information. However, it is difficult for UAVs to fly at the canyon bottom due to the problems of a lack of signal and occlusion in the slot canyon, which also similarly causes the problem of local data shadows.

To obtain high-resolution terrain datasets of slot canyons, considering the uniqueness and complexity of geomorphological forms of slot canyons, such as variable slopes and partial vegetation cover, comprehensive data through an integration scheme is necessary to reduce local data shadows generated by a single data acquisition method. The data integration scheme has been applied in cultural heritage [22], geomorphology research [23], and environmental monitoring [24]. Data integration has also been applied to complex landscapes, including terraces and alpine terrains in challenging areas [13,25]. In slot canyon systems, the more complex and unique topography and vegetation cover certainly complicate the application of data acquisition and integration. Therefore, this paper proposes an integrated workflow using fine UAV photogrammetry and BMLS techniques and tests the availability of the data integration scheme for point cloud data from multiple sources in slot canyon geomorphological analysis. The main contributions of this work to the high-resolution terrain reconstruction of slot canyons are as follows:

(1) The fine flight of UAVs based on a rough model of the large area, which can avoid collision with obstacles such as powerlines or flying into restricted areas, allowing users to perform UAV photogrammetry faster, safer, more detailed, and with higher quality.

(2)    This integration scheme significantly reduces data shadows in the 3D point clouds produced solely by a single mapping technique, providing a means for 3D mapping of extremely steep slopes and overhangs frequently present in rugged terrain.

(3)    This study can provide a high-fidelity terrain dataset with sufficient spatial detail, including a complete point cloud and its derivatives such as DEM and 3D mesh, which can be used for quantitative analysis of the morphological evolution and genesis of the slot canyons.

## 2. Materials and Methods

An integrated scheme using UAV photogrammetric data with BMLS data provides an accurate survey of the slot canyon, producing detailed DEM and 3D data. For effective data integration, several aspects need to be considered, especially the distribution of targets in the UAV photogrammetric data and BMLS data, which are crucial for data co-registration in the final integration. For this reason, it is important to develop a standard workflow that can address all the possible issues, such as co-registration issues and the errors of the DEM generation process. The following sections will provide all the details about this scheme, including data acquisition, data processing, data registration and integration, and DEM and mesh generation (Figure 1).

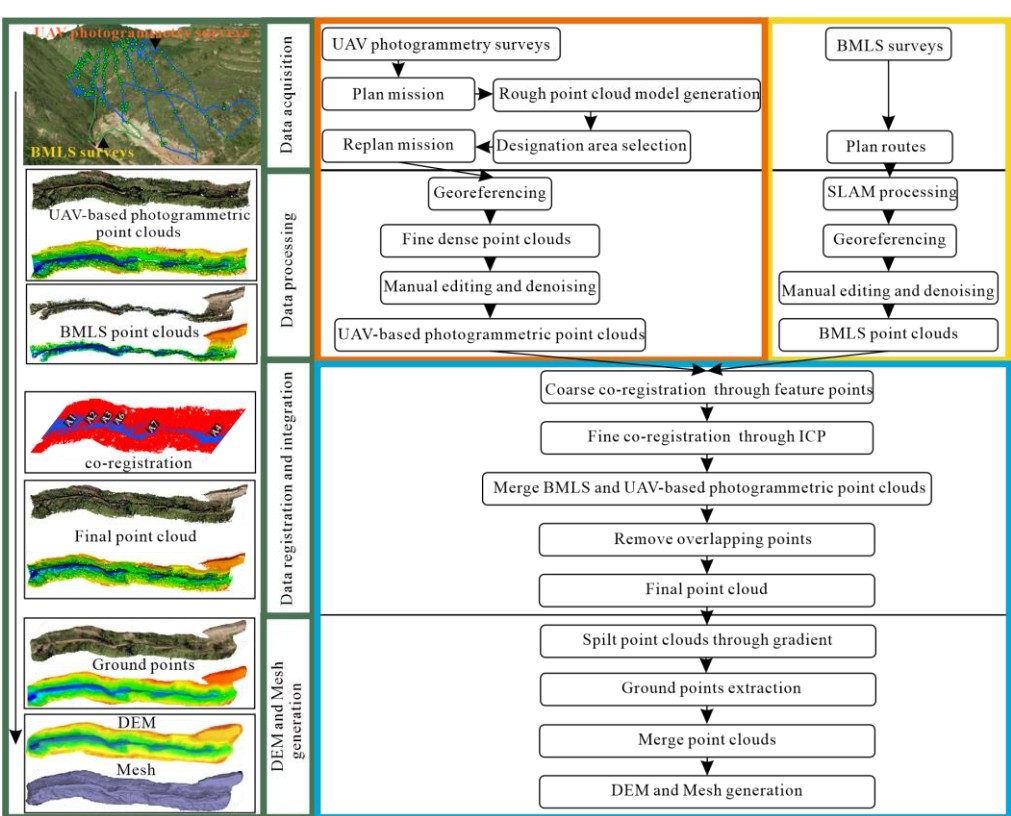

**Figure 1.** The proposed scheme of integration of UAV photogrammetric data and BMLS data.

### 2.1. Study Area

The study area is in Zhidan County, Yan'an City, Shaanxi Province, China (36°47′01″ N, 108°49′58″ E; 36°47′06″ N, 108°49′46″ E). It is tectonically located in the Ordos Block, bounded by a peripheral fault system, and in a state of uplift with weak tectonic activity (Figure 2a). The nearly horizontal strata are the Early Cretaceous Luohe Formation, composed of yellow, violet, and red aeolian sandstones, in which the deformation structure is not developed. There are no major fracture structures. Rather, it only has a few joints. The Maoxiang slot canyon is twelve kilometers long, several tens of meters deep, and mostly one to two meters wide, has extremely variable terrain slopes, and lies on the south and

west sides of the watershed between the Beiluo River system and the Yanhe River system (Figure 2b,c). A segment of the Maoxiang slot canyon was selected as the test site for this study (Figure 2d). The vegetation mainly develops in the loess-covered area at the top of the canyon, while vegetation cover in bedrock canyons is sparse. Notably, the canyon is a typical one-line sky landscape viewed from above (Figure 2e,f).

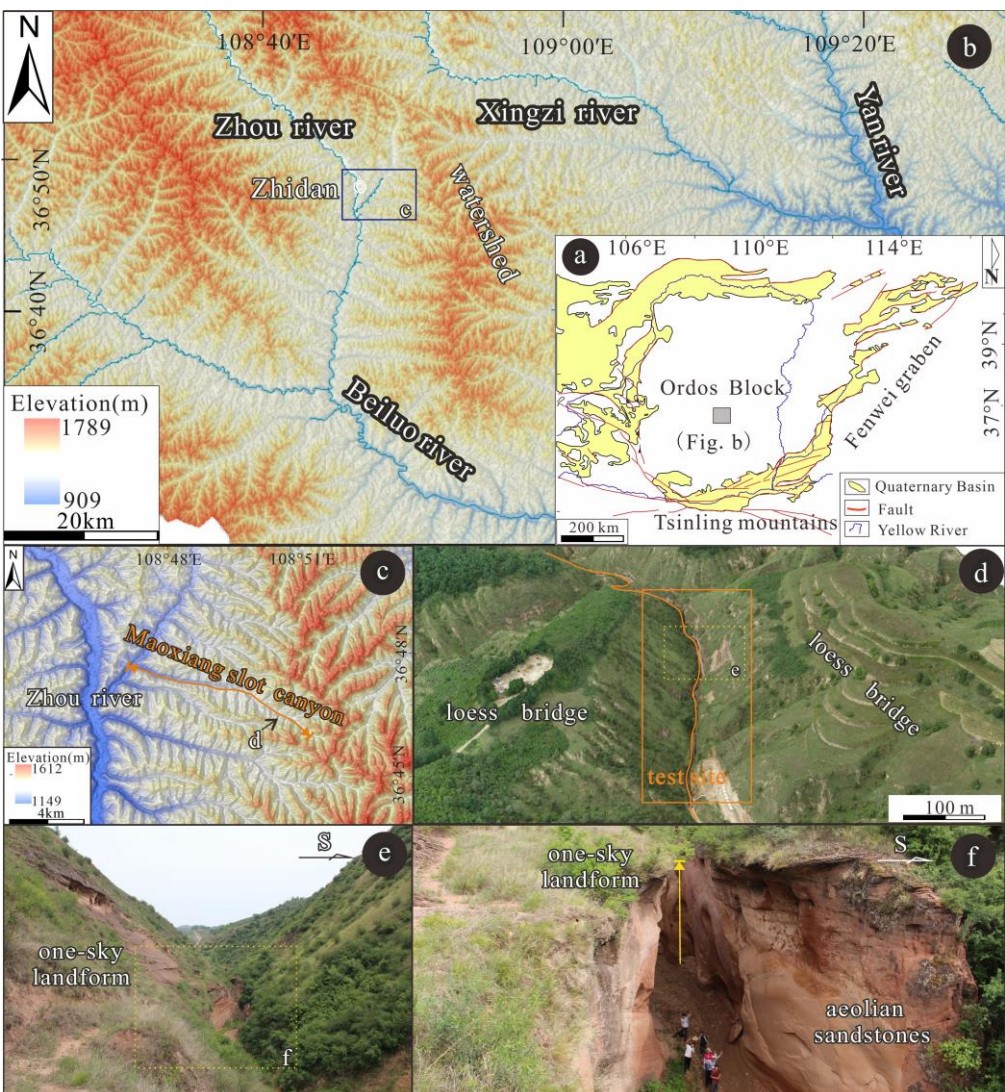

**Figure 2.** Study area. (**a**) tectonic map of the study area (modified by the literature [26]); (**b**,**c**) schematic diagram of DEM upstream of the Maoxiang slot canyon (from ALOS 12.5 m DEM, below contained in search.asf.alaska.edu); (**d**) orthophoto acquired by this study; (**e**,**f**) field photos representing the one-sky landform.

## 2.2. Data Acquisition and Processing

In this study, a range of measurement methods was integrated that enabled us to create high-resolution topographic data of the slot canyon. The results of UAV photogrammetry, which contain more textural information than aerial scanning results, can help us to preliminarily determine lithology for geomorphological and geological research, such as revealing the genesis of the canyon. Low-altitude aerial photogrammetry using a UAV was used to obtain the point clouds to express the topographical details at the canyon top. Compared to TLS, BMLS was used to acquire point clouds to reflect the topographical characteristics of the canyon bottom due to its portability, fast and strong adaptability.

In addition, Ground Control Points (GCPs) and Ground Control Planes (GCPLs) were collected to test the accuracy of the data.

### 2.2.1. UAV Photogrammetry

Images were collected with a 20 million-pixel FC6310R HD camera attached to a three-axis aerial gimbal mounted on a DJI Phantom 4 RTK. As rugged terrain brings some limitations such as occluded areas with considerable height differences and complex geometry (e.g., data shadows), the coverage and reliability of UAV surveys are confronted with major challenges [27]. In particular, these limitations are more significant due to the deep and narrow characteristics of slot canyons, which complicates further processing [28]. Given the difficulty of data acquisition in complex terrain, fine flight path creation was necessary to ensure complete data acquisition through automatic flight. Many 3D reconstruction softwares such as the Metashape® software package and DJI Terra® and mission planning software such as DJI Pilot® and Mission Planner® can be used to achieve fine flight path creation. The DJI Pilot® app and the Metashape® software package by Agisoft version 1.6.2 were used for flight path creation in this study. In the fieldwork, two flight missions were used for data acquisition: the first mission was mainly used to obtain a large range of data, and the second mission was planned to create a fine flight path based on the data obtained from the first mission. The flights were performed on-site between 10:00 and 16:00 on a sunny day. The images captured from the first flight mission were imported into Metashape® installed on the portable laptop to generate a rough point cloud model using the lowest quality settings. The second flight mission was generated by accurately reselecting the areas, including the area of interest, the safety area, the powerline area, and the obstacle area, to generate a route with a higher ground spatial resolution (GSD) (Figure 3). The automatically generated Keyhole Markup Language (KML) files of the fine flight mission were imported into the DJI Pilot® to accomplish the task of generating a detailed model. The basic information on UAV photogrammetry acquisitions is illustrated in Table S1.

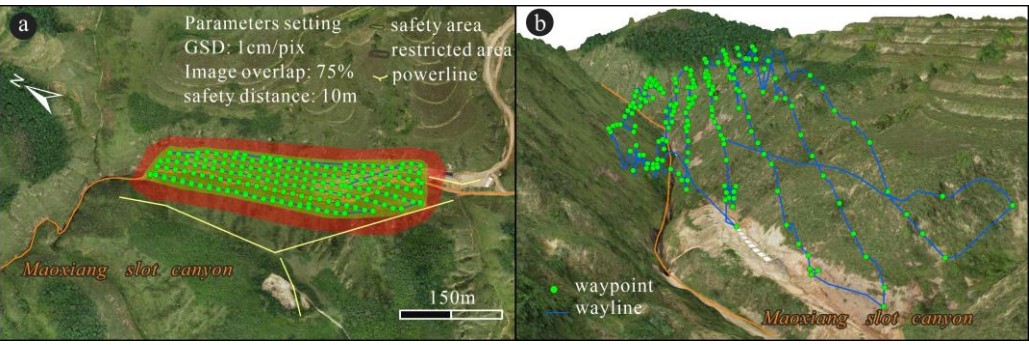

**Figure 3.** Fine planning path map of UAV. (**a**) the vertical view; (**b**) the stereo view.

The detailed model processing was performed in Metashape® according to the process of previous studies [17,18]. The reconstruction software was run on a high-performance computer (Table S1). All images have been estimated for image quality. Quality issues are mainly related to water surface reflections and lighting changes during flight [20]. The light conditions change with the changing altitude of the sun during the flight. A subset of images was selected to create the final point clouds for these quality considerations. The parameter settings of the main steps are illustrated in Table S1.

### 2.2.2. Backpack Mobile Laser Scanning

The LiBackpack DGC50 backpack LiDAR system (Digital Green Valley Technology Co., Ltd., Beijing, China) equipped with two Velodyne Puck VLP-16 LiDAR sensors rotating in the horizontal and vertical directions, high-precision GNSS, and a panoramic camera with a resolution of 4320 × 2160 pixels, 600,000 laser pulses per second for emission with a



vertical field angle of ±90° and a horizontal field angle of 0–360°, were used to collect data at the canyon bottom. The maximum measurement range of this system could reach 100 m, the relative accuracy of the data is within 3 cm, and the absolute accuracy is no more than 5 cm. However, the relative accuracy is high when the optimal scanning distance is within 50 m [29]. It provides an absolute scale and relative orientation, information that can be converted to real-world coordinates by setting the position and orientation of equipment and GCPs prior to data acquisition (Figure 4a). During field data acquisition, a surveyor zigzagged along the canyon, circling the GCPLs as much as possible or pausing for a while to subsequently identify them (Figure 4b). The walking route should end in an open area with stable GNSS signals outside the sample area to ensure that the point clouds contain geographic information, as shown in Figure 4a. Notably, the sparse version of the collected data can be previewed in real-time, making it easy to adjust the acquisition path to avoid acquisition omissions. After data acquisition, the original point clouds and GNSS station logs were imported into the LiFuser-BP® software (version 1.4) for matching. Meanwhile, the point clouds were retrieved, collected, and processed with the geographic information and trajectory of the site.

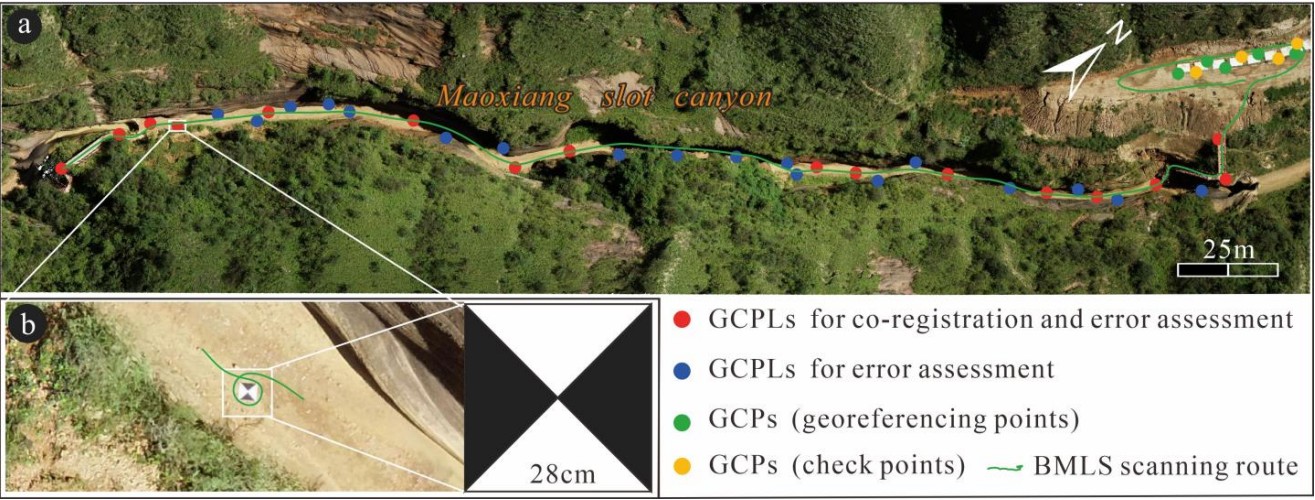

**Figure 4.** GCPLs and GCPs. (**a**) spatial distribution of GCPLs and GCPs; (**b**) the GCPLs.

A total of nine points were measured on the buildings. Five GCPs were used to georeference BMLS point clouds and four GCPs could be used as checkpoints to evaluate georeferenced accuracy, the distribution of which is shown in Figure 4a. This study uses HI-TARGET RTK TS5 to measure the locations of GCPs in the open area. Its horizontal positioning accuracy is $\pm (8 + 1 \times 10^{-6}$ D) mm, the vertical accuracy is $\pm (15 + 1 \times 10^{-6}$ D) mm, the static positioning horizontal accuracy is $\pm (2.5 + 0.5 \times 10^{-6}$ D) mm, and the vertical accuracy is $\pm (5 + 0.5 \times 10^{-6}$ D) mm, where D is the distance between the measuring points. The ten times smoothing method was used for collecting the location and the average value was recorded. The GCPLs with an "intersect" pattern (28 cm × 28 cm) were placed as identifiable targets throughout the scene (Figure 4b). In order to map the attitude of each target, a hand-held compass was used to occupy the center of each GCPL. A network of thirty (30) GCPLs was established to facilitate BMLS and UAV photogrammetry surveys (Figure 4a). Twelve GCPLs were collected to serve as data registration and error assessment due to the common visibility in the BMLS and UAV surveys and other GCPLs were only used for error assessment due to visibility in the BMLS surveys. In addition, the attitudes of the four red staircases were measured at higher elevations throughout the site for better vertical control within the point clouds.

### 2.3. Co-Registration and Data Fusion

Both independently acquired datasets contained local data shadows or sparse coverage that could be reduced by the fusion of the BMLS and UAV photogrammetric point clouds. Co-registration is required before fusion for aligning different point clouds. In this study, the co-registration of the point clouds in the LiDAR360® software (version 5.0) consists of manually selecting feature points for coarse registration and running an iterative closest point (ICP) algorithm for fine registration based on obtaining the initial values. The ICP algorithm proposed by Besl and Mckay is one of the most classical point cloud registration algorithms [30]. The basic principle of the algorithm is to find the corresponding points in the reference point cloud and the alignment point cloud according to the nearest point method, and form two sets of corresponding point sets P {$p_1$, $p_2$, ... , $p_i$, ... , $p_n$}, Q {$q_1$, $q_2$, ... , $q_i$, ... , $q_n$}. Then, the rigid body transformation (rotation matrix R and translation vector T) is solved according to the correspondence relationship, so that the objective function shown in Equation (1) gets the minimum value. The rigid body transformation is applied to the alignment point clouds, and the process is continuously iterated until a specific convergence criterion is met.

$$f(R, \, T) = \frac{1}{N_P} \sum_{i=1}^{N_P} ||q_i - (R \times q_i + T)||^2 \tag{1}$$

Using the Alignment tool to match point pairs manually is very useful for aligning point clouds precisely. This tool allows the user to align two point clouds by picking at least three matching point pairs in the two point clouds [31]. The error contribution of each point pair to matching point pairs can be seen at the time of selection (e.g., drop again and choose the worst pair). New points can be added to two sets at any time (even after pressing the align button) to add more constraints and get more reliable results. The coarse registration was performed using 16 locations (Figure 5a,b), and the cropped point cloud was filtered manually. In this study, only 14 locations with similar transformations were selected, and the final transformation matrix was calculated as the average of the observations at each location.

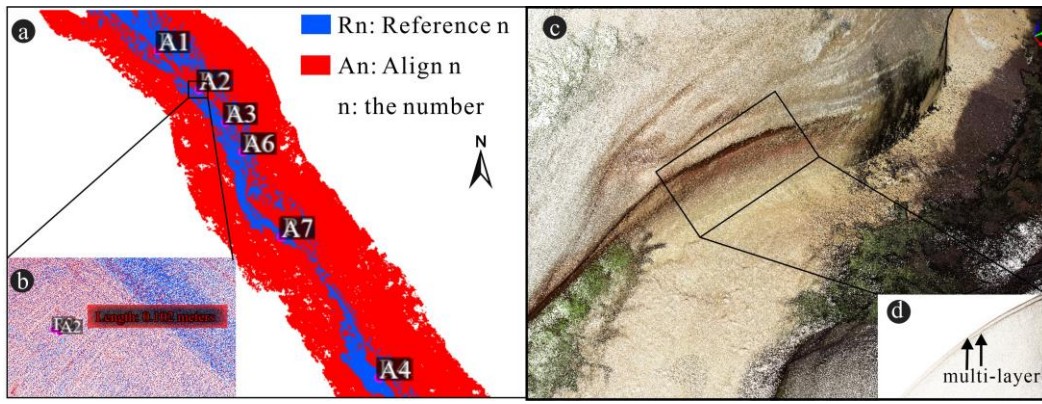

**Figure 5.** Co-registration. (**a**,**b**) same name point selection; (**c**,**d**) multi-layer point cloud.

After coarse registration, the ICP algorithm was used for fine registration. Three key parameters were taken for the ICP algorithm: (1) Root Mean Square (RMS) difference, which controlled the quality of the result. The minimum RMS improvement between two consecutive iterations was set to 0.00001. (2) Number of iterations, which controlled the computation time. The maximum number of steps for the algorithm registration computation was set to 30. (3) Random sampling points, which were set to 50,000. Above this limit, clouds were randomly resampled at each iteration. Notably, overlapping points had to be removed, or an iso-density refinement should be required to ensure the effectiveness of the

3D modeling since the high density of the point clouds in the overlapping areas, or due to multi-layer as the presence of registration errors (Figure 5c,d).

### 2.4. Ground Points Extraction and DEM Generation

DEM, as a continuous representation of the surface relief, can be generated by point clouds of a discrete digital representation [32]. In this study, the final point clouds used for generating DEM were the ground point clouds which remained after filtering out off-ground points such as vegetation and buildings. A variety of filter algorithms with specific geometry criteria to eliminate off-ground points have been developed so far [33–36]. For example, three parameters were taken for eliminating off-ground points: (1) the terrain angle, which limits the maximum slope of the surface to be created, (2) the iteration angle, which limits the angular range between the point to be classified and the known ground points, and (3) the iteration distance, which limits the range of distances between the point to be classified and the surface to be created. The iteration angle and iteration distance can be adjusted to be larger in areas with large terrain fluctuations [32]. However, the current filtering algorithms were not adaptive and not suitable for all terrain changes.

Due to the variable terrain slopes of the study area, the point clouds were first performed with gradient extraction and divided into three levels of slope: 0–30°, 30–60°, and 60–90° (Figure 6a,b). In addition, previous experiments have proved that the average total error of the improved progressive TIN densification (IPTD) algorithm was lower than 3.15%, and the kappa coefficient can reach 89.53%. The type I error (error of incorrectly dividing ground points into off-ground points) was about 1–8%, and the type II error (error of incorrectly dividing off-ground points into ground points) was generally 1–7% [34]. The IPTD algorithm can meet the requirements of this ground point extraction [36]. The steps of the IPTD algorithm included the following three aspects: (1) Obtain the potential ground seed points using the morphological open operation (Figure 6c,d). (2) Filter potential ground seed points to obtain accurate ground seed points (Figure 6e). (3) Densify using an iterative densification TIN method to extract the ground points (Figure 6f). The IPTD algorithm was applied to the point cloud extracted according to the slope, and different parameters were adjusted respectively to extract the ground points (Table 1). The IPTD filtering results were manually revised more than once to ensure accuracy. Finally, the ground points with different slopes were integrated to obtain the ground point results (Figure 6g,h).

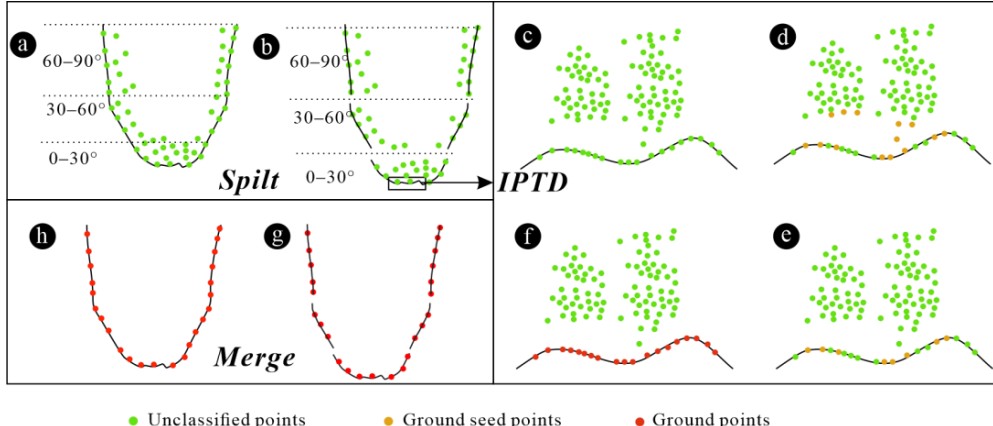

**Figure 6.** Ground point extraction. (**a**) raw data; (**b**) split data; (**c–f**) main steps in IPTD (modified by the literature [34]). (**c**) the raw data of portion; (**d**) potential ground seed points identified using the opening operation; (**e**) accurate ground seed points after eliminating the off-ground points; (**f**) extracted ground points using progressive TIN densification; (**g,h**) the results of ground points.

**Table 1.** The parameters of ground point extraction.

| Slope (°) | Terrain Angle (°) | Iteration Angle (°) | Iteration Distance (m) |
|---|---|---|---|
| 0–30 | 30 | 25 | 0.5 |
| 30–60 | 60 | 55 | 0.8 |
| 60–90 | 90 | 75 | 1.2 |

Through LiDAR360® software, point cloud smoothing was performed based on ground points, and a smooth continuous DEM was generated using the irregular triangle interpolation method. The final DEM interpolation was performed on a square grid with a grid cell size of 0.05 m, which corresponds to the density of the final spatially homogenized point clouds. Furthermore, due to the DEM data structure of the canyon, the slope information of the canyon was not represented well in this study. A polyhedral mesh was generated using the Poisson reconstruction plugin in the CloudCompare® software to supplement the sidewall morphology [37].

## 3. Result and Analysis

### 3.1. The Generation of Point Clouds and DEM

Two point clouds were generated by UAV photogrammetry. The rough model obtained during the first flight at an altitude of about 130 m was only used to plan the detailed flight mission. The orthophoto is shown as the base map in Figure 3a. The fine UAV-based photogrammetric point clouds were obtained for the second time with a GSD of 1 cm/pixel at an altitude of 20–35 m, which appear in RGB color in Figures 7a and 8a. As shown in Figure 8a,b, the point density ranged from 0 to 6000 points per square meter, and high-density point cloud areas were concentrated at the top of the model with sporadic distribution at the bottom, while low-density areas were scattered throughout the model. In addition, more gaps in the lower part are presented in the blue rectangle (Figure 8a).

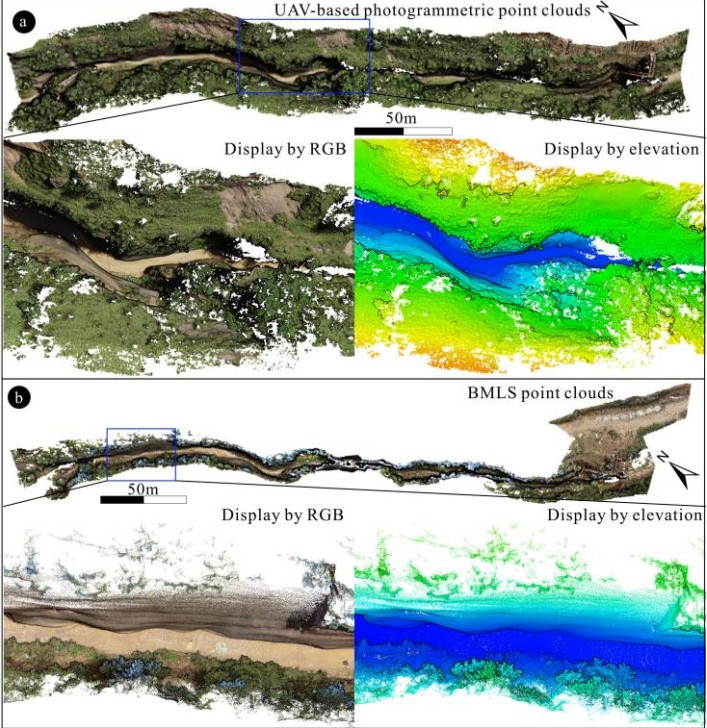

**Figure 7.** Point clouds of Maoxiang slot canyon. (**a**) UAV-based photogrammetric point clouds. The diagram below is a partially enlarged view, the left is displayed with RGB color, and the right is displayed by elevation; (**b**) BMLS point clouds, the diagram below is a partially enlarged view, the left is displayed with RGB color, and the right is displayed by elevation.

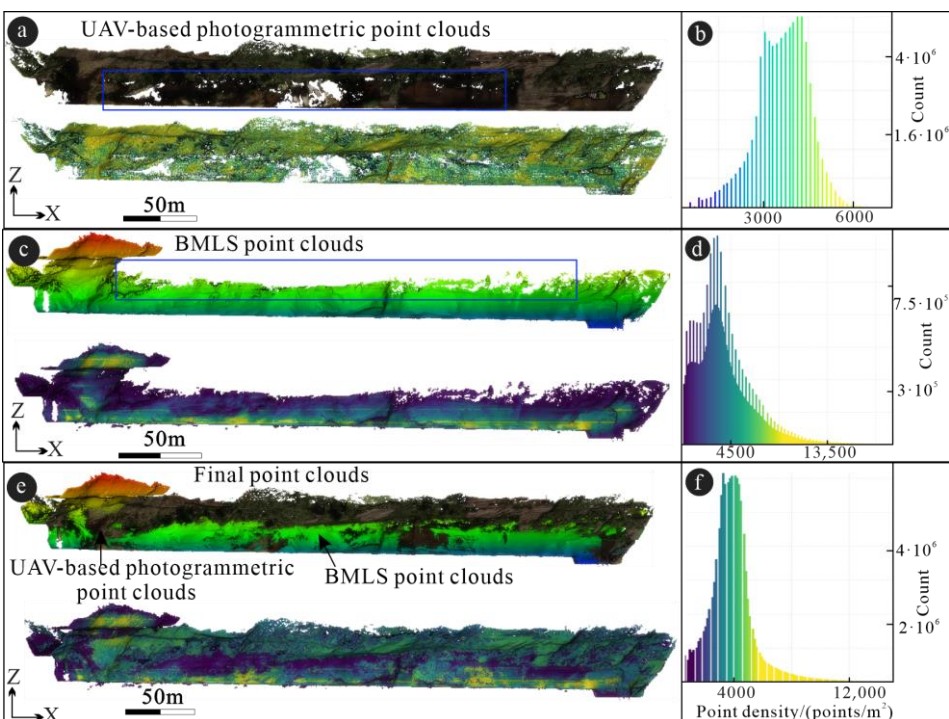

**Figure 8.** Point cloud models and their point density of the Maoxiang slot canyon. (**a**) Above the RGB color display of the UAV-based photogrammetric point clouds and below the point density distribution; (**b**) the point density distribution histogram of the UAV-based photogrammetric point clouds; (**c**) above is the RGB color representation of the BMLS point clouds, and below is the point density distribution; (**d**) the point density distribution histogram of the BMLS point clouds; (**e**) top is the RGB color display of the final point clouds, and the bottom is the point density distribution; (**f**) the point density distribution histogram of the final point clouds.

The BMLS survey generated more than 37 million points over a distance of 300 m. Figures 7b and 8c display the results of the BMLS point clouds. Figure 8d displays the point density of the BMLS point clouds. The yellow area is the scanning position of the equipment at the canyon bottom, which is essentially distributed in line with a density of 13,500 points/m². The point density decreases from the center of the yellow area to both sides, suggesting a negative correlation between point density and scanning distance. Generally, there is no missing data at the bottom and sidewalls of the canyon, but obtaining information about the top at an effective scanning distance is difficult due to the rough terrain. In this study, the final point clouds contained all data from the canyon top and bottom without gaps, based on the integration of UAV-based photogrammetric point clouds and BMLS point clouds (Figure 8e). In addition, the point density of the complementary parts remained unchanged, and the point density in the overlapping area increased after integrating and de-overlapping (Figure 8e,f).

The DEM was generated based on the extracted ground points using the irregular triangular network interpolation method, which showed a height difference of forty meters in the study area with a spatial resolution of five centimeters (Figure 9a). It was worth noting that the ground resolution of the acquired photos reached the millimeter level. The point density of the BMLS point cloud was higher than that of the photogrammetric point cloud, so it was feasible to generate the centimeter-level pixel resolution model based on these data. From the color of the elevation display, the four elevation layers can be seen clearly in a very narrow range. The area was mostly rugged and steep terrain composed of numerous concave and vertical surfaces resulted in sharp changes in slope and altitude, which limits the representation of the terrain of slot canyons by DEM. Therefore, a 3D mesh model was generated to characterize the shape of the inner sidewall of the canyon

(Figure 9b). The lower part of Figure 9b is an enlarged view of a portion of the cropped model. The left part is the solid model and the right part is the wireframe model, which can not only clearly express the topographic fluctuation changes, but also represent the irregular triangular network structure of the reconstructed model well. The DEM and mesh data complemented each other to characterize the 3D terrain of the canyon, allowing for a detailed analysis of the canyon.

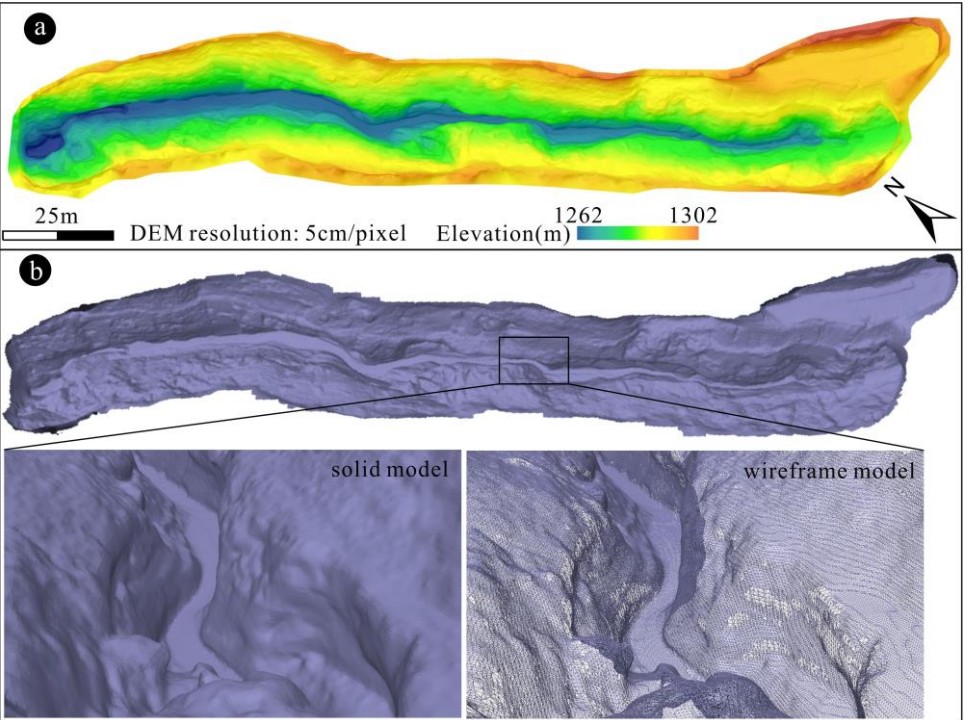

**Figure 9.** DEM and mesh of Maoxiang slot canyon. (**a**) DEM; (**b**) mesh, the below is a partially enlarged view, the left is a solid model, and the right is a wireframe model.

### 3.2. Error Assessment

In this paper, the qualitative and quantitative accuracy assessment was performed in three key steps of the workflow to verify the rationality and feasibility of the approach.

(1) The results of the ground point extraction. Many published papers related to filtering algorithms adopted a manually revised method of generating reference data to analyze the accuracy of algorithms. Therefore, this study adopted the revised classification results as reference data [34,38,39]. The cross-matrix of errors, including type I error, type II error, total error, and kappa coefficient, was computed to evaluate filtering performance, as shown in Table 2. The specific cross-matrix information is shown in Table S2. The type I error of the 0–60 terrain slope range was larger than the type II error because the average vegetation cover of these areas was high. As a result, several misclassified ground points gave a high type I error. In an area with 60–90 slopes and sparse vegetation cover, the type II error value was higher than the type I error. The reasons can be explained as follows: Most of the data in these areas were collected by the BMLS. BMLS points were mostly ground points, and a few misclassified off-ground points would result in a high type II error. All kappa coefficient values were greater than 0.80, indicating good classified results.

**Table 2.** The accuracy in filtered results.

| Slope (°) | Type I Error (%) | Type II Error (%) | Total Error (%) | Kappa Coefficient |
| --- | --- | --- | --- | --- |
| 0–30 | 3.34 | 1.40 | 4.74 | 0.89 |
| 30–60 | 5.82 | 0.60 | 6.42 | 0.82 |
| 60–90 | 1.77 | 7.36 | 9.13 | 0.80 |

In addition, the point clouds before and after vegetation removal are displayed in Figure 10a. The comparison results show that the irregular protrusions (vegetation) were eliminated and mainly concentrated on the green parts. The filter results were qualitatively analyzed by creating multiple point cloud profiles before and after filtering, stepping through the data, and looking at a small portion of the terrain at a time. Five 1-m-wide cross sections were evenly intercepted on the model at 50 m intervals to present the significant difference in terrain laterally before and after vegetation filtering due to the elimination of irregular protrusions (vegetation) (Figure 10b).

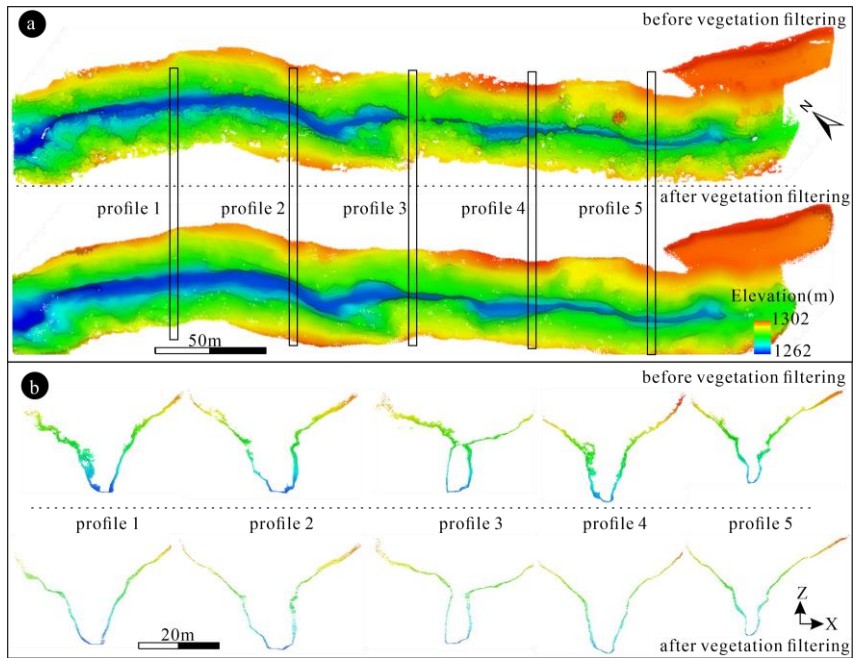

**Figure 10.** The results of ground points extraction. (**a**) point clouds before and after vegetation removal; (**b**) example cross-section used for vegetation removal.

(2) The error of attitude and size. Point clouds have various sources of error, so the error and accuracy in the final result must be identified [40]. The resulting BMLS clouds were georeferenced, with five GCPs achieving the coordinate transformation standard deviation of 35.5 mm. The accuracy of the standard deviation of the four checkpoints was 31.4 mm. The UAV-based photogrammetric point clouds were georeferenced using WGS84 coordinates (EPSG: 4326) measured by the UAV's internal RTK-GNSS antenna and recorded in the EXIF information of the imagery. In addition, there were no GCPs acquired for such an accuracy assessment of the UAV-based photogrammetric point clouds because it was impractical in rugged terrain such as slot canyons to place the GCP targets in the area of interest and measure their location with GNSS, which would require additional workforce and more time. Table 3 contains only the accuracy of the internal RTK positioning information provided by Metashape® under an absence of GCPs' points. Only the aerial measurement was used. However, it is important to note that accuracy statistics reported in Metashape® software are often overly optimistic when used as an indicator of overall point cloud accuracy [41], so relative accuracy needs to be considered. This calculation usually uses the difference between the actual object in the field and models to describe the value. The relative error distribution was calculated by comparing the size and attitude of GCPLs in the BMLS point clouds and UAV-based photogrammetry point clouds with the field measurements. The GCPLs and red staircases were selected in this experiment as measurement objects to estimate the model error (Figure 11). The specific data are shown in Table S2.

**Table 3.** Average camera location error.

| Mission | X (cm) | Y (cm) | Z (cm) | Total (cm) | RMS Reprojection (Pixel) |
|---|---|---|---|---|---|
| Flight mission 1 | 0.56 | 0.54 | 1.03 | 1.29 | 0.54 |
| Flight mission 2 | 0.48 | 0.50 | 1.05 | 1.27 | 0.59 |

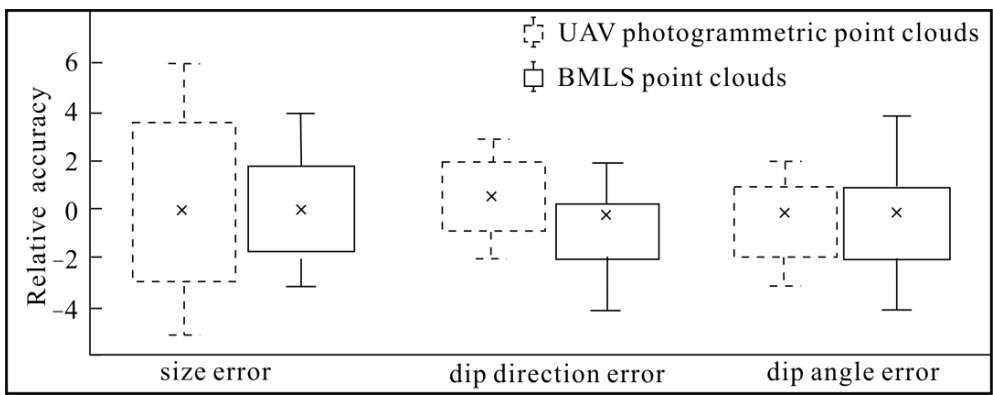

**Figure 11.** The relative accuracy of the UAV-based photogrammetric point clouds and BMLS point clouds. Note: The dimension of the size error is mm; the dimension of the dip direction error and dip angle error is °.

The size error was measured from 12 GCPLs and the attitude error was described by 19 and 34 groups of attitudes of GCPLs in the UAV-based photogrammetric point clouds and BMLS point clouds, respectively. For the attitude error, the measurement of UAV-based photogrammetric data and BMLS data includes 15 GCPLs and four staircases, as well as 30 GCPLs and four staircases, respectively. The size error of the UAV-based photogrammetric point clouds ((−5 mm)–(+6 mm)) is slightly larger than that of the BMLS point clouds ((−3 mm)−(−4 mm)) because the distance to obtain data was farther, or the image-based reconstruction compared to direct acquisition of the laser pulse could result in greater distortion (Figure 11a). The relative error of the dip direction of the UAV-based photogrammetric point clouds is (−2°)–(+3°), and the dip angle error is (−3°)–(+2°). The dip direction error of the BMLS point clouds is (−4°)–(+2°), and the dip angle error is (−4°)–(+4°) (Figure 11b,c). In general, the attitude error in the dip direction or dip angle of the UAV-based photogrammetric data is better than that of the BMLS data. Combined with the field acquisition environment, the formation of large errors in BMLS data could be related to the lack of a good closed-loop optimization of the acquisition route.

(3) The registration error of point clouds. Root Mean Squared Error (RMSE) represents the deviation between the observed and actual values, which is often considered an important indicator of registration results [42]. In particular, this registration procedure assumed that the BMLS data was internally registered (mutually oriented) with a higher accuracy than the UAV-based photogrammetric data. Therefore, the BMLS point clouds were applied as the reference entities, and the UAV-based photogrammetric point clouds were applied as the alignment entities. Based on our data, the coarse registration error of 12 points was 0.29 m and the fine registration error was 0.028 m, and there was no stratification or offset after fine registration.

The initial UAV photogrammetric point clouds, the UAV photogrammetric point clouds of coarse registration, and the UAV photogrammetric point clouds of fine registration were loaded into the CloudCompare® software and compared to the BMLS point cloud using the M3C2 distance algorithm tool [43]. The cloud-to-cloud comparisons undertaken with the M3C2 algorithm showed that fine registration point clouds produced a lower error distribution (Figure 12a,c,e). The significant error is mainly distributed in the area of the sidewall and the dense vegetation cover. With sparse vegetation cover and flat areas such as the valley bottom, the error was negligible (Figure 12b,d,f).

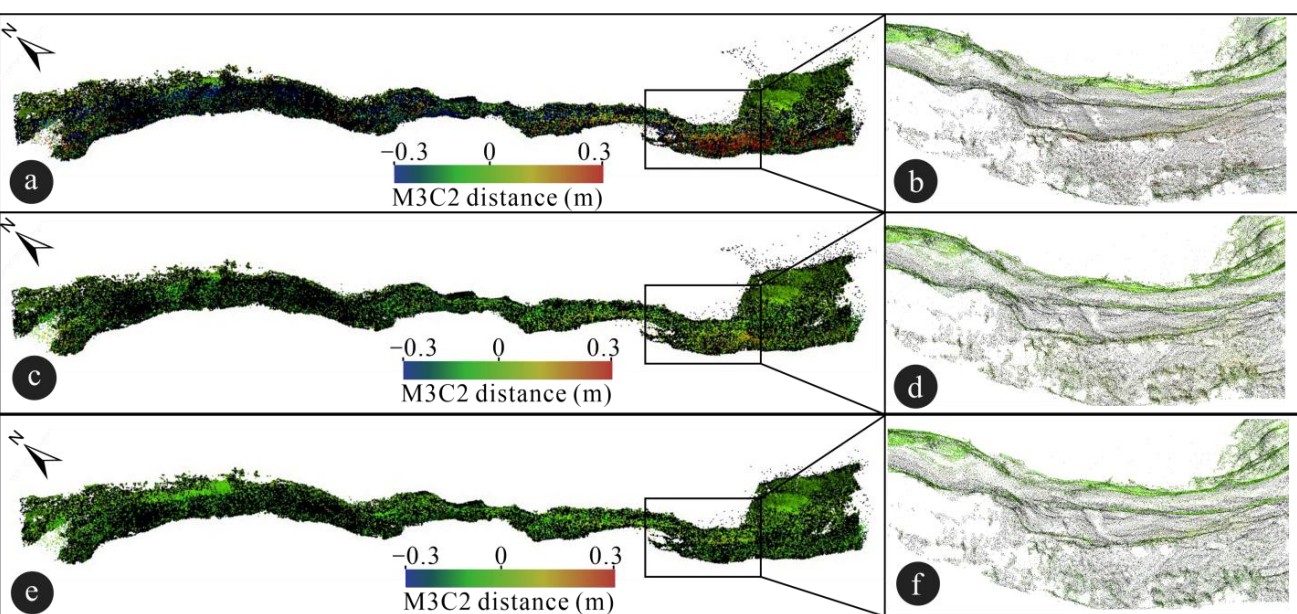

**Figure 12.** Comparison of data quality with the M3C2 algorithm of Cloudcompare® software by using the BMLS point clouds as reference. (**a**) initial point clouds; (**b**) the partially enlarged view of initial point clouds; (**c**) coarse registration point clouds; (**d**) the partially enlarged view of coarse registration point clouds; (**e**) fine registration point clouds; (**f**) the partially enlarged view of fine registration point clouds.

## 4. Discussions

### 4.1. Complete Data Acquisition and Solutions

Given the uniqueness and complexity of the geomorphological forms of slot canyons, such as rugged topography and the occurrence of vegetation, applying a single data acquisition method to obtain complete data in this environment was challenging. Therefore, a comprehensive data acquisition scheme was tested in this slot canyon (Figure 13). The data acquisition is divided into two parts: top data acquisition and bottom data acquisition. The canyon bottom is in a no satellite signal, short-visibility environment. The BMLS survey is used to obtain the bottom data through zigzagging along the canyon. While the data outside the top steep terrain area is obtained using UAV photogrammetry.

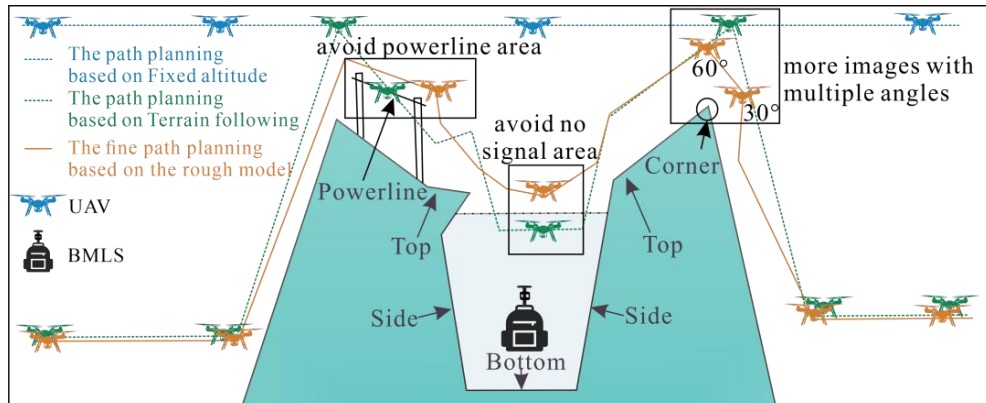

**Figure 13.** Comprehensive data acquisition scheme and the comparison of flight path planning methods.

Effective and safe path planning for UAVs was necessary to obtain the image data for the 3D reconstruction. As shown in Figure 13, the path planning based on Fixed-altitude selects the study area from a 2D satellite map without considering the impact of the actual

terrain and the latest data. At high altitude areas, data coverage (image) is reduced due to the decrease in relative altitude, and even the 3D reconstruction will fail due to the low overlap [44]. Path planning based on terrain following can overcome this difficulty. It can adapt to different terrain according to DEM data. Variable elevation routes can be automatically generated to keep the GSD consistent and obtain high-quality data [45]. However, it still cannot avoid the potential safety problems of flying into the no-signal area or colliding with obstacles such as thin powerlines which cannot be detected by the flight obstacle avoidance system.

Meanwhile, the shooting angle is fixed in both path plans, and corner areas cannot be mapped precisely due to the lack of data coverage. Therefore, this study developed a workflow to create the UAV fine flight path based on rough model of a large area. Fine data acquisition was achieved through the use of twice low-altitude flights. The flight path generated was a combination of multiple waypoints, and each waypoint had its elevation and angle, allowing users to perform UAV photogrammetry faster, safer, in more detail, and with higher quality.

The steps required for the fine flight of UAVs based on a rough model of the large area are as follows: The first survey of GCPs with RTK-GNSS was necessary; second, follow the operational area to take flight photos and build a high-level model of the point clouds that requires little detailed knowledge of data acquisition. The accuracy (alignment step) and quality (dense point clouds step) parameters were set to Low and Lowest, respectively, to ensure data integrity. Then, data shadow areas were also selected in the rough model and fine flight missions were created. For fine path planning, it is necessary to mark the safe and restricted areas and set the shooting parameters such as GSD and image overlap. The automatically generated KML files of the fine flight mission should be imported into DJI Pilot® or other flight path creation software to call the drone to perform a flight mission and conduct faster, safer, more detailed, and higher quality data acquisition. The workflow of fine flight path creation is shown in Figure 14.

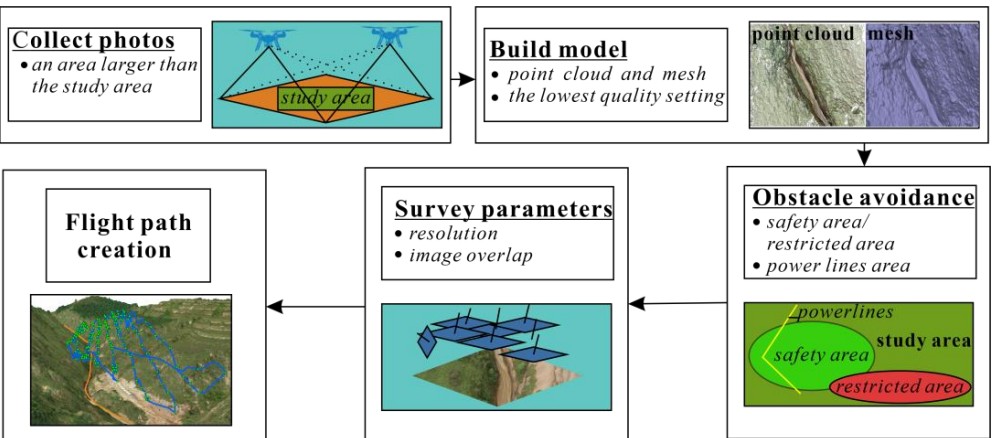

**Figure 14.** The workflow of fine flight path creation.

*4.2. The Applicability Analysis*

4.2.1. The Feasibility of the Integrated Method

Traditional single-point measurements still take a long time to collect sufficient data density to obtain detailed and complete field scene data. They often provide limited, fragmented data and insights, and tend to generate data with widely varying spatial densities and uncertainties [46]. Meanwhile, DEM data from satellites cannot effectively identify the interior of the canyon because of the signal occlusion from small shrubs on either side of the canyon. The satellite observation intends to produce orthophotos, so the side wall generally cannot be observed well [47]. Their resolution of tens of meters to sub-meters scales can only describe the large and medium landforms, but cannot describe small and micro landscapes, such as terraces, potholes, ledges, faults, and fractures. UAV

photogrammetric surveys provide a reliable, fast, and inexpensive method to obtain terrain data with high relief complexity and operational limitations [18]. However, there are still many challenges in slot canyon terrain reconstruction, mainly manifested in the following two aspects: (1) UAVs hardly get any data on the long concave surface formed by the sandstone potholes in the slot canyon, as in Figures 8a and 15a. The sidewalls at the bottom of the canyon are mainly empty and no data are available. (2) Setting up the GCP is difficult. There is no signal inside the canyon, and the terrain conditions on either side do not allow surveyors to reach reliable deployment checkpoints [48]. Although BMLS surveys can penetrate vegetation cover such as grass and dwarf conifers [49], they are also limited by scan distance and accessibility as follows: (1) The point density is negatively correlated with the scan distance, meaning that point density decreases as scanning distance increases [50]. (2) Since the acquisition of point cloud data was limited by the occlusion of the viewing angle, which included the rich BMLS data shadows caused by curved surfaces and boulders, and the inclined view of the ground platform, it is difficult to obtain top information [51]. As shown in Figures 7b and 15b, the top and bottom data contrast enormously compared to the UAV-based photogrammetric point clouds. The bottom data are essentially non-missing, and the top has no data. As data areas moved away from the scan line, the density was constant and continuously decreased.

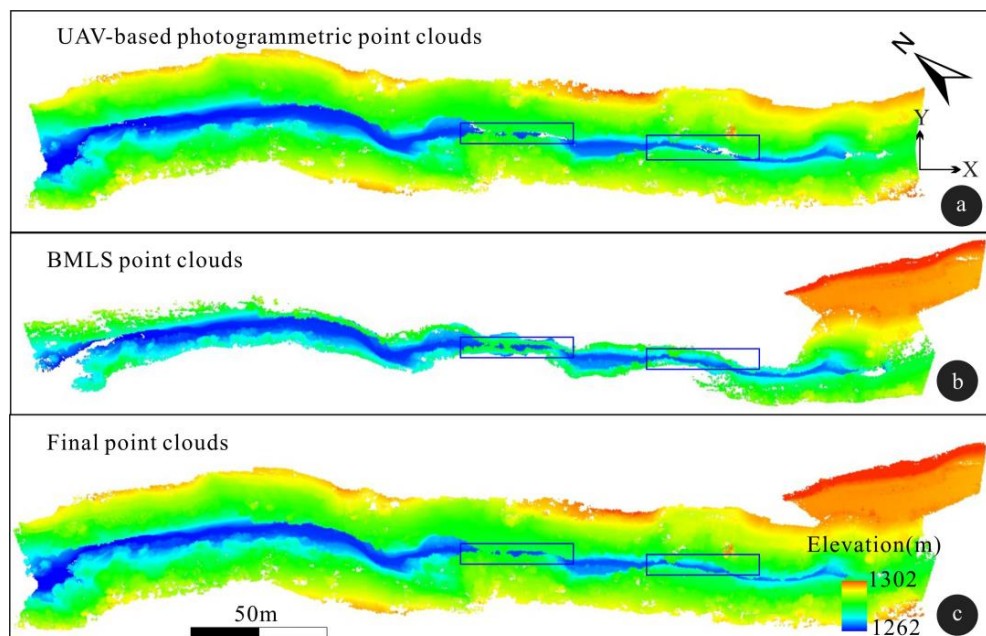

**Figure 15.** The vertical view of point cloud models. (**a**) UAV-based photogrammetric point clouds; (**b**) BMLS point clouds; (**c**) final point clouds.

Therefore, an integrated scheme of remote sensing techniques, including BMLS, RTK GNSS, and UAV photogrammetry, is proposed to solve these problems. These techniques produce data sets composed of irregularly spaced (X, Y, Z) spatial coordinates or point clouds [52]. Previous studies have shown that the georeferencing of UAVs equipped with accurate GNSS equipment and operating in RTK can maintain reasonable accuracy without GCPs measurements [53,54]. However, this work would lead to a lack of GCPs to check the data accuracy; GCPLs were added for relative accuracy evaluation. For BMLS georeferencing, GCPs were chosen to be deployed in an open area, combined with the point clouds with relative orientation and absolute scale calculated by the SLAM algorithm [55]. In Figures 8e and 15c, the integration of BMLS data and UAV photogrammetry data formed an effective complementarity and the integration of multi-source data made up for the shortage of single acquisition method and obtained complete slot canyon point clouds. Generally, point cloud smoothing is required to provide a more uniform spatial

distribution of point clouds, which is important for creating DEM with spatially consistent quality through spatial interpolation [56]. In the example, a minimum distance of 0.05 m was considered sufficient to represent the terrain, generating the corresponding DEM. Furthermore, the resulting dense point clouds and 3D mesh are valuable and unique for canyon studies in terms of geometric accuracy and complementary application [15].

### 4.2.2. The Capability of Data Applications

Conventional geomorphological surveys, including length, width, slope, and slope variation, are used to describe slot canyon landforms [57]. DEM, a common data source for topography, is widely used in the application of large-scale flat terrain, representing various surface forms such as cliffs and rough surfaces formed by rubble [10]. High-resolution DEM created by fusing BMLS data and UAV-based photogrammetric data can identify and analyze a wide range of specific landforms and geomorphological processes, opening new possibilities for multi-scale landscape modeling and segmentation to represent a hierarchy of topographic and surface processes. However, the DEM data structure is limited when simulating the apparent verticality of the surface in rough terrain [25]. Although the point clouds recorded the 3D point coordinates of this shape, DEM uses interpolation to average multiple heights, which can only represent a single vertical height value, seriously distorting the actual terrain [58]. The solution for analyzing such specific topographical features is to use original point clouds or to model the surface as a 3D mesh. Therefore, the point clouds and mesh were combined to help analyze these elevations to compensate for the lack of the single elevation value of the surface represented in a single DEM pixel. There are some differences in terrain profile between DEM and point clouds. Black straight lines in Figure 16a represent the locations of profiles 1 and 3. It can be seen that the profile shapes created by profile 1 from DEM and point clouds were consistent in the open canyon area (Figure 16b). Among them, the profile shapes of DEM were relatively smooth, while the profile shapes of the point clouds reflected more terrain details. Instead, in profile 3 of the canyon area with a narrow top and a wide bottom, representing the actual shape of the terrain by point clouds was more accurate than DEM (Figure 16b). In addition, combined with Figure 10b, the canyon development was considered at the V-shaped valley stage, and the bottom of slot canyons became narrower towards the southeast, indicating the trend of retrogressive erosion.

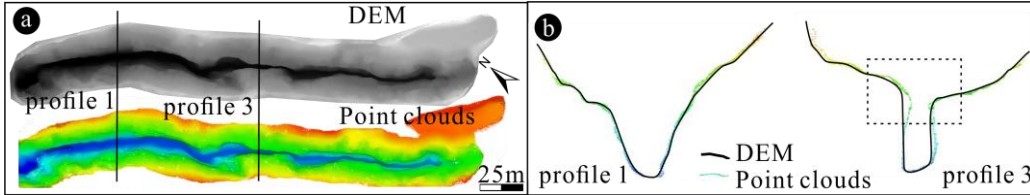

**Figure 16.** Terrain differences between DEM and point cloud. (**a**) the location of terrain profiles; (**b**) example profiles for terrain differences.

Figure 17 compares the range of values distribution of the derived slope between the DEM surface and the 3D mesh surface. The corresponding position in the DEM of Figure 17a can be shown in more detail and comprehensively from another view of the mesh model of Figure 17b. Particularly, it can be seen in Figure 17c that the proportion of DEM data is significantly larger than that of mesh data when the slope is less than 30 degrees. Conversely, the proportion of DEM data is significantly smaller if the slope is greater than 30 degrees. The difference between the DEM data and the mesh data becomes increasingly larger with the increase of the slope, indicating that the 3D mesh data reflect more information to a certain extent supplemented by the information of the DEM data to accurately reflect the actual scene.

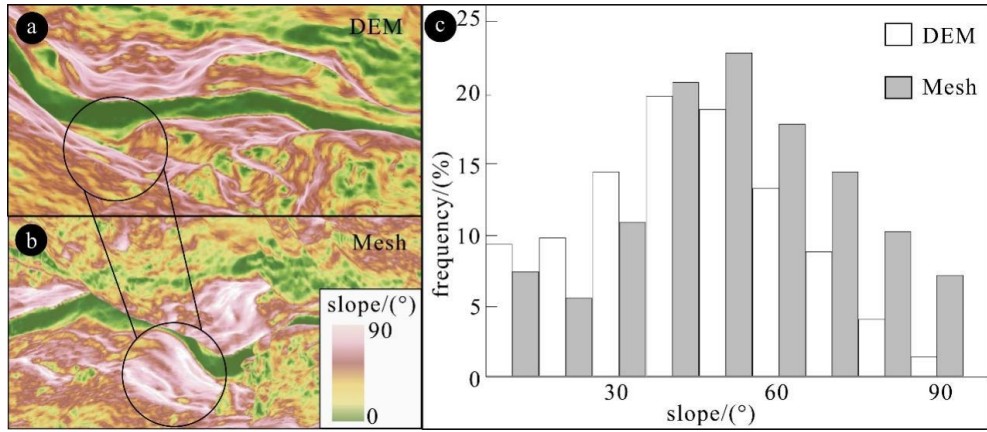

**Figure 17.** The slope maps. (**a**) the slope map of DEM; (**b**) the slope map of Mesh; (**c**) the slope distribution histogram.

In addition, 3D data can also help to understand both some 2D data and the real world [59]. For example, 3D point clouds can be used for trajectory roaming to experience the real environment in the field (Figure 18a). It can also be used to split strata and create histograms based on 3D point clouds or mesh models (Figure 18b,c), interpret geological features, measure attitudes of strata, compare attitudes of strata and canyons, and determine the relationship between canyon development, evolution, and structure [52,60]. The histogram of the corresponding profile of eolian sandstone in this area is shown on the left side of Figure 18c. On the right side of Figure 18c, the strike measured values of the 24 oblique beddings in the canyon area shown in Figure 18c and the seven measured values of the overall strike of the canyon were described. The specific data measured from the model are shown in Table S3. The strike of the canyon was similar to the strike of the rock formations in this area and the angle between them was only 16°, indicating the development of the slot canyon was controlled by strata structures.

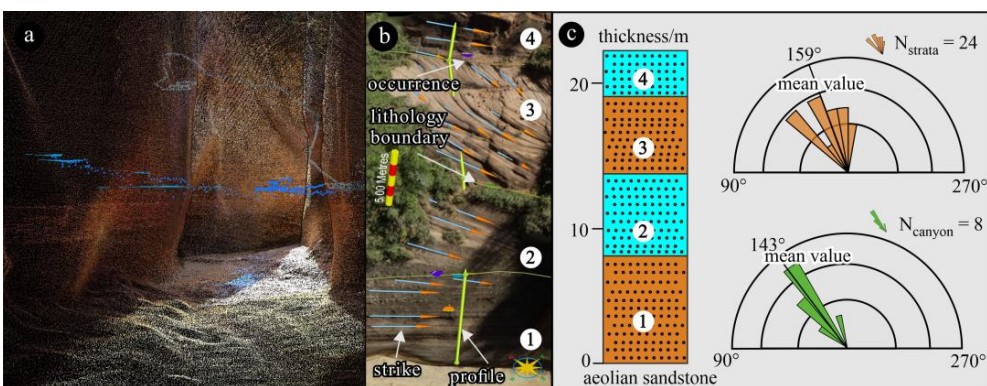

**Figure 18.** The application of 3D point clouds and mesh. (**a**) roaming on a 3D point cloud (the blue and gray dots represent the roaming trajectory); (**b**) geologic interpretation and measurement in the 3D mesh; (**c**) the histogram of strata and the rose diagram of the strike of strata and canyon.

## 5. Conclusions

This study has highlighted how data integration can be used to create highly accurate terrain datasets (centimeter-level) in the context of complex topography with local no satellite signal area and rugged terrain, such as in slot canyon systems.

(1) The integration scheme with BMLS and UAV photogrammetry applied in slot canyon systems has proven to be an effective solution for terrain reconstruction in a more accurate, complete, and realistic way. In addition, multi-scale observation and planning are necessary to ensure the safety and efficiency of collecting data. Compared

with path planning based on Fixed-altitude methods and path planning based on Terrain following, the fine flight of UAVs based on a rough model of the large area can avoid collision with obstacles such as powerlines or flying into the restricted area, allowing users to perform UAV photogrammetry faster, safer, in more detail and with higher quality.

(2) The registration process plays a key role in the accurate data integration of the point clouds. The accuracy of the registration by BMLS and UAV photogrammetric point clouds in this study is good with a RMSE of 0.028 m and there are no point clouds with stratification and offset. In addition, the vegetation filtering results of splitting the integrated point clouds into different slope segments used for ground point extraction are good, with all kappa coefficient values greater than 0.80.

(3) The high-resolution terrain dataset achieved by data integration and includes a complete color point cloud, DEM and mesh, are starting points for generating valuable geometric parameters such as slope factor and for interpreting geologic features such as the attitude and thickness of sedimentary beddings about the slot canyon system. They provide a useful supplement for revealing the morphological evolution and genesis of slot canyons.

**Supplementary Materials:** The following supporting information can be downloaded at: https://www.mdpi.com/article/10.3390/drones6120429/s1, Table S1: Computers, Main characteristics of UAV Photogrammetric data acquisitions, and Settings used in Agisoft Metashape software (version 1.6.2); Table S2: The cross-matrix information of classification results; Table S3: The relative error of attitude and size of UAV-based photogrammetric data and BMLS data; Table S4: The geometry data extracted from the mesh model.

**Author Contributions:** Conceptualization, Y.X. and R.W.; methodology, Y.X.; software, Y.X.; validation, Y.X., R.W. and Z.X.; formal analysis, X.W. (Xi Wang); investigation, Y.X. and X.W. (Xingwei Wang); resources, X.W. (Xi Wang); data curation, X.W. (Xingwei Wang); writing—original draft preparation, Y.X. and X.W. (Xi Wang); writing—review and editing, Y.X., R.W., X.W. (Xi Wang), Z.X. and J.L.; visualization, Y.X.; supervision, R.W.; project administration, R.W.; funding acquisition, R.W. All authors have read and agreed to the published version of the manuscript.

**Funding:** This research was funded by the National Natural Science Foundation of China, grant number 42271014.

**Institutional Review Board Statement:** Not applicable.

**Informed Consent Statement:** Not applicable.

**Data Availability Statement:** Not applicable.

**Acknowledgments:** The authors extend their appreciation to the National Natural Science Foundation of China for funding this research work through the project.

**Conflicts of Interest:** The authors declare no conflict of interest. The funders had no role in the design of the study; in the collection, analyses, or interpretation of data; in the writing of the manuscript; or in the decision to publish the results.

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
