# Peer review of "High-Resolution Terrain Reconstruction of Slot Canyon Using Backpack Mobile Laser Scanning and UAV Photogrammetry"

_drones, doi:10.3390/drones6120429_

Round 1

Reviewer 1 Report (Previous Reviewer 3)

Dear authors,

Thanks for the effort in considering the comments. Generally, the paper still needs revision. Also, I want to note that any reviewer comments or questions should reflect on the manuscript, and only responding/arguing with the reviewer would not be enough as the reviewer is a sample of your prospective readers. If he has faced ambiguity, it might also be the case for the reader.

Please find attached the comments.

Success!

Author Response

Response to Reviewer 1 Comments

Point 1: Dear authors,

Thanks for the effort in considering the comments. Generally, the paper still needs revision. Also, I want to note that any reviewer comments or questions should reflect on the manuscript, and only responding/arguing with the reviewer would not be enough as the reviewer is a sample of your prospective readers. If he has faced ambiguity, it might also be the case for the reader.

Please find attached the comments.

Success!

Response 1: We gratefully appreciate your affirmation of our modification and give us more valuable suggestions. It is very important. In this review, reviewer comments or questions have been reflected on the manuscript as requested and the manuscript is amended in the abstract, introduction, discussion and Conclusion. Other suggestions are also modified as requested. Meanwhile, we have carefully checked the manuscript and made some English changes. This detailed point-to-point response is listed below.

Point 2: I strongly suggest to revise the manuscript in case of parts a "novel approach" is proposed. This is more like integration solution to get advantage of different solutions. This novelty is not clear yet. A combined method? but not a novel approach. This novelty is not clear yet. A combined method? but not a novel approach

Response 2: Thank you for your comment. We have realized this problem and have used the term “integrated scheme” to express it in the full text.

Point 3: No explanation is added to the text?! If something is reviewers question, it might be other readers too

Response 3: Thank you for your comment. We have revised it. It was modified as requested:

“Such high-resolution integration terrain datasets reduce local data shadows produced solely by individual datasets, providing a starting point for revealing the morphological evolution and genesis of slot canyons.”

Point 4: still too much descriptive than being scientific (order of several, significant, good)

Response 4: Thank you for the comment. This is our lack of precise statement. We have replaced these vague expressions with more scientific terms. It was modified as follows:

“Data integration of BMLS and UAV photogrammetry can obtain accurate terrain datasets, with a Root Mean Squared Error (RMSE) of point cloud registration of 0.028m. Such high-resolution integration terrain datasets reduce local data shadows produced solely by individual datasets, providing a starting point for revealing the morphological evolution and genesis of slot canyons.

Although such methods can provide cm- or mm-level accuracy and precision for measuring single points, they often provide limited, fragmented data and tend to generate data with widely varying spatial densities and uncertainties, while collecting data dense enough to generate useful terrain data would take a long time [9].”

Point 5: newly developed 3D point cloud technology

These are not point cloud technology, these are techniques and instruments, not for point cloud, but for point cloud generation

Response 5: Yes, as requested, we have rewritten this sentence. The modification is listed as follows:

“Recent advances in geospatial data collection, especially the application of active laser scanning and passive photogrammetry-based methods have greatly improved the ability for high-resolution terrain research [13,14,15].”

References:

  1. Li, L.; Wang, R.; Lin, J.; Xiao, Z.; Hui, Y. A novel approach for extraction of ripple mark parameters based on SfM. Sediment. Geol. 2019, 392, 105523.
  2. Chudá, J.; Hunčaga, M.; Tuček, J.; Mokroš, M. The handheld mobile laser scanners as a tool for accurate positioning under forest canopy. The International Archives of Photogrammetry, Remote Sensing and Spatial Information Sciences. 2020, 43, 211-8.
  3. Cucchiaro, S.; Fallu, D.J.; Zhang, H.; Walsh, K.; Van Oost, K.; Brown, A.G.; Tarolli, P. Multiplatform-SfM and TLS Data Fusion for Monitoring Agricultural Terraces in Complex Topographic and Landcover Conditions. Remote Sensing. 2020, 12, 1946.

Point 6: LIDAR is an active instrument while SFM is a processing algorithm. The logic of drafting this part is completely wrong. you can compare active methods with passive photogrammetry based methods but not Lidar with SFM, you mixed up. some are techniques,some are instruments,some

are just algorithms, The same problem still exists

Response 6: Yes, as requested. We 're sorry we couldn’t answer your question effectively last time. We 've revised this paragraph. The modification is listed as follows:

“Recent advances in geospatial data collection, especially the application of active laser scanning and passive photogrammetry-based methods have greatly improved the ability for high-resolution terrain research [13,14,15]. Laser scanning, as a non-contact active measurement method, is in the ability to sample several kinds of surfaces (e.g., top of vegetation canopy, inter-canopy surfaces, and ground) that are in the line of sight of the laser beam until impermeable surface restrains further penetration of the laser energy [16]. Photogrammetry-based methods also have been increasingly applicated in the geosciences field due to their low cost and high efficiency [17-20]. In addition to obtaining the point clouds with spatial coordinates (X, Y, Z), photogrammetry data also provide a large amount of texture information that can distinguish the objects. Restricted lines of sight severely in slot canyons impede the use of the widely used Terrestrial Laser Scanning (TLS) technique, and data collection is spatially limited due to its static nature, requiring multiple adjustments to achieve full coverage [9]. Surveys using Backpack Mobile Laser Scanning (BMLS) or Hand-held Laser Scanning (HLS) typically by sophisticated Simultaneous Localization and Mapping (SLAM) algorithms that avoid the problem of multiple adjustments of TLS. They can capture information on vertical surfaces (e.g., cliffs, steep slopes, landslides) at the canyon bottom with no satellite signal environments [14, 21], but rarely obtain sample boundaries across the upper zones within the field of view (e.g., rock shelters, building roofs), limiting the acquisition of complete data. Aerial laser scanning and aerial photogrammetry can obtain the upper data through UAV systems. Compare with aerial laser scanning, the results of UAV photogrammetry contain more texture information. However, it is difficult for UAVs to fly at the canyon bottom due to the problems of no signal and occlusion in the slot canyon, which also presents the problem of local data shadows similarly.”

Point 7: line 91: Still not convince why this methodology is innovative, needed and valuable

Response 7: Thank you for the generous comment. As requested, we have rewritten this section. The modifications are listed as follows:

“1) The fine flight of UAVs based on a rough model of the large area can avoid collision with obstacles such as powerlines or flying into a restricted area, allowing users to perform UAV photogrammetry faster, safer, more detailed and with higher quality.

2) This integration scheme greatly reduces data shadows present in the 3D point clouds produced solely by a single mapping technique, providing means for 3D mapping of extremely steep slopes and overhangs frequently present in rugged terrain. 

3) This study can provide a high-fidelity terrain dataset with sufficient spatial detail, including a complete point cloud and its derivatives such as DEM and 3D mesh, which can be used for quantitative analysis of the morphological evolution and genesis of the slot canyons.”

Point 8: Why imaging, not aerial scanning? explanation needs to be reflected also on the paper

Response 8: Thank you for the generous comment. We have modified and expanded the description of this problem. The modifications are listed as follows:

Section 2.2: “In this study, a range of measurement methods was integrated that enabled us to create high-resolution topographic data of the slot canyon. The results of UAV photogrammetry containing more texture information than the aerial scanning results can help us to preliminarily determine lithology for geomorphologic and geological research, such as revealing the genesis of the canyon. Low-altitude aerial photogrammetry using a UAV was used to obtain the point clouds to express the topographical details at the canyon top. Compared to TLS, BMLS was used to acquire point clouds to reflect the topographical characteristics of the canyon bottom and side wall due to its portability, fast and strong adaptability. In addition, Ground Control Points (GCPs) and Ground Control Planes (GCPLs) were collected to test the accuracy of the data.”

Point 9: where is it?

Response 9: Thank you for the generous comment. This is our negligence. It has been marked red in Table S1.

Point 10: Geo-referenced or Geo-referencing?

Response 10: Thank you for the generous comment. It is a mistake. We have revised this mistake as requested.

Section 2.2.2: “Five GCPs were used for georeferencing BMLS point clouds and four GCPs could be used as checkpoints to evaluate georeferenced accuracy,”

Point 11: Figure 11: line 385: Figures are really hard to understand and can be presented in simpler way. I can not recognize any changes

Response 11: Thank you for the generous comment. And we have modified the figure layout and the figure as requested:

Figure 11. The relative accuracy of the UAV-based photogrammetric point clouds and BMLS point clouds. Note: The dimension of the size error is mm; the dimension of the dip direction error and dip angle error is °.

Reviewer 2 Report (Previous Reviewer 2)

Thank you for the modification and the implementation of images, now the article is much more clear

Author Response

Response to Reviewer 2 Comments

Point 1: Thank you for the modification and the implementation of images, now the article is much more clear

Response 1: Thank you very much for your affirmation of our article. We will make persistent efforts in this field and strive to do more meaningful things in the future.

Reviewer 3 Report (New Reviewer)

The paper documents an exercise in using an integrated approach using multiple capture modalities in a challenging terrain environment. The paper is generally well presented and sufficient detail is provided for those who may need to replicate the results. The contents are highly relevant to those working in the field of remote sensing.

Perhaps the only test I think would have been valuable would be to perform ground truth measurement comparisons, I would be skeptical about the cm resolution accuracy claim.

Author Response

Response to Reviewer 3 Comments

Point 1: The paper documents an exercise in using an integrated approach using multiple capture modalities in a challenging terrain environment. The paper is generally well presented and sufficient detail is provided for those who may need to replicate the results. The contents are highly relevant to those working in the field of remote sensing.

Perhaps the only test I think would have been valuable would be to perform ground truth measurement comparisons, I would be skeptical about the cm resolution accuracy claim.

Respond 1: Thank you very much for your affirmation of our article. We will make persistent efforts in this field and strive to do more meaningful things in the future. The ground resolution of the acquired photos has reached the millimeter level, so it is feasible to generate the centimeter-level pixel resolution model based on these photos. In addition, the point density of the lidar point cloud is higher than that of the photogrammetric point cloud, so it can also generate the cm-level resolution model. Therefore, the integrated model can achieve the cm level resolution.

“Section 3.1: It was worth noting that the ground resolution of the acquired photos reached the millimeter level, the point density of the lidar point cloud was higher than that of the photogrammetric point cloud, so it was feasible to generate the centimeter-level pixel resolution model based on these data.”

Reviewer 4 Report (New Reviewer)

25/11/2022

Dear authors,

In the manuscript High-Resolution Terrain Reconstruction of Slot Canyon Using Backpack Mobile Laser Scanning and UAV Photogrammetry you proposes an integrated approach and workflow using RTK GNSS, BMLS, and UAV photogrammetry for generating a comprehensive, integrated, centimetre-resolution slot canyon terrain dataset, including a complete point cloud and its derivatives such as Digital Elevation Model (DEM), mesh data. The approach is demonstrated on a sandstone development area in the Ordos Block.

General comments

The theme is interesting; however, these are more or less known facts. Based on that, I think this is professional work, not scientific. No novelties is brought, just use of a wide range of data.

The abstract is full of imprecise statements. After the research, words like: 'could effectively' or 'several centimetres’ cannot be used. You must accurately interpret your results and highlight them in the Abstract.

The content of the Introduction is not satisfactory. You list a lot of facts in Introduction without references. The text is confused and not systematized. It is not entirely clear what your method yielded.

In the second chapter, which you should call Materials and Methods, you need to describe the procedure more clearly and what the purpose of all this is. Because all that could be done differently and by other methods.

The Conclusion contains a part of the text that does not belong there. It is not clear whether the results are general rules for anyone using UAV in photogrammetry or are related only to the tests performed. New findings have not been adequately highlighted and discussed. There is no comparison of results with other methods. In this way, the method cannot be valorized.

Specific comments (are in the manuscript)

-          Line 12-15 - This sentence is imprecise and untrue. Even before these technologies, experts studied slot canyons. Compared to previous times, today's technology makes it easier and gives better results. I guess that's what you wanted to point out.

-          Line 19-20 - This is not a precise statement and cannot be expressed in this way after conducting research. You either proved something or you didn't, or you achieved better results with your method, or you didn't. This is a scientific paper and the results must be interpreted precisely.

-          Line 58 - What are they? Please provide references.

-          Line 88-89 - Another not precise and not entirely accurate statement. This statement has no place in the Introduction (if you do not provide references), but in the Conclusion with a detailed interpretation of the results.

-          Line 100-101, 103 - Is this a scientific contribution? Do you think no one else has done it before you?

-          Line 119 - Such type of manuscript should be written in the 3rd person.

-          Line 144 - Do you think you are really the first to do something like this? Also, you cite references after this claim!?

-          Line 362 - Which? Please provide references.

-          Line 584-589 - This text has no place in the Conclusion. The facts from the previous text are repeated here.

Best regards

Author Response

Response to Reviewer 4 Comments

Point 1: The theme is interesting; however, these are more or less known facts. Based on that, I think this is professional work, not scientific. No novelties is brought, just use of a wide range of data.

Response 1:  We gratefully appreciate your valuable suggestions. After conscientious consideration, we have revised the paper and prepared this detailed point-to-point response listed below. The manuscript is amended in the abstract, introduction, material and method, and conclusion. More precise statements are used, and the procedure and the purpose of all are more clearly described. We also have carefully checked the manuscript and made some English changes.

We have modified the statement. This study integrated the advantages of various methods to make a useful application in this slot canyon, which provides an integrated solution for canyon measurement. Sufficient detail is provided for those who may need to replicate the results. And this study provides a terrain dataset with high fidelity and spatial detail, including a complete point cloud and its derivatives such as DEM and mesh to enhance the use of quantitative analyses of land-surface topography. The dataset would provide a starting point for revealing morphological evolution and the genesis of the slot canyons.

Point 2: The abstract is full of imprecise statements. After the research, words like: 'could effectively' or 'several centimetres’ cannot be used. You must accurately interpret your results and highlight them in the Abstract.

Response 2: Thank you for your comment. Precise statements are added to accurately interpret our results and highlight them in the abstract. And We also have checked the full text and similar questions have been revised.

“Data integration of BMLS and UAV photogrammetry can obtain accurate terrain datasets, with a Root Mean Squared Error (RMSE) of point cloud registration of 0.028m. Such high-resolution integration terrain datasets reduce local data shadows produced solely by individual datasets, providing a starting point for revealing the morphological evolution and genesis of slot canyons.”

Point 3: The content of the Introduction is not satisfactory. You list a lot of facts in Introduction without references. The text is confused and not systematized. It is not entirely clear what your method yielded.

Response 3: Thank you for your comment. The content of the Introduction is reorganized to make it clear and systematic. And eight references are added to the corresponding places.

“They have unique morphological characteristics that are manifested in three aspects as follows [5]: 1) Narrow at the bottom and sometimes wide at the top. The wide top of slot canyons reaches tens of meters but the bottom width is nearly tens of centimeters, allowing only one person to pass. 2) Steep sidewalls. Variable slopes and a large number of high-angle planes are great characteristics of slot canyons. 3) Partial vegetation cover. At the top, more Quaternary deposits are remained because of the gentle slope, supporting the vegetation development, while at the bottom, more bedrock is exposed owing to the steep slope, leading to sparse vegetation. Such a unique landform represents the characteristics of connecting valleys with gorges [6], which mostly present the valley-in-valley landform and one-sky landscape, such as Antelope Canyon in the Colorado Plateau, USA, and slots in China red beds landscapes. They have an important scientific and tourism value. Additionally, the slot canyons also exhibit many dynamic geomorphological phenomena associated with mass movements such as landslides and erosion that often alter the morphology of the terrain surface [7].

There is also much research on large-scale terrain and geomorphology analysis using DEM data from satellites [10,11].

Recent advances in geospatial data collection, especially the application of active laser scanning and passive photogrammetry-based methods have greatly improved the ability for high-resolution terrain research [13,14,15].

References:

  1. Young, R.; Young, A. Erosional Forms. In: Young R, Young A, eds. Sandstone Landforms. Berlin, Heidelberg: Springer Berlin Heidelberg, 1992: 81-99.
  2. Sanders, D.; Wischounig, L.; Gruber, A.; Ostermann, M. Inner gorge–slot canyon system produced by repeated stream incision (eastern Alps): Significance for development of bedrock canyons. Geomorphology. 2014, 214, 465-84.
  3. Watkins, C.M.; Rogers, J.D. A New Look at Landslides of the Vermilion and Echo Cliffs, Northern Arizona. Environ. Eng. Geosci. 2022, 28, 173-92.
  4. Ibrahim, M.; Al-Mashaqbah, A.; Koch, B.; Datta, P. An evaluation of available digital elevation models (DEMs) for geomorphological feature analysis. Environ. Earth Sci. 2020, 79, 1-11.
  5. Koukouvelas, I.K.; Zygouri, V.; Nikolakopoulos, K.; Verroios, S. Treatise on the tectonic geomorphology of active faults: The significance of using a universal digital elevation model. J. Struct. Geol. 2018, 116, 241-52.
  6. Li, L.; Wang, R.; Lin, J.; Xiao, Z.; Hui, Y. A novel approach for extraction of ripple mark parameters based on SfM. Sediment. Geol. 2019, 392, 105523.
  7. Chudá, J.; Hunčaga, M.; Tuček, J.; Mokroš, M. The handheld mobile laser scanners as a tool for accurate positioning under forest canopy. The International Archives of Photogrammetry, Remote Sensing and Spatial Information Sciences. 2020, 43, 211-8.
  8. Cucchiaro, S.; Fallu, D.J.; Zhang, H.; Walsh, K.; Van Oost, K.; Brown, A.G.; Tarolli, P. Multiplatform-SfM and TLS Data Fusion for Monitoring Agricultural Terraces in Complex Topographic and Landcover Conditions. Remote Sensing. 2020, 12, 1946.

Point 4: In the second chapter, which you should call Materials and Methods, you need to describe the procedure more clearly and what the purpose of all this is. Because all that could be done differently and by other methods.

Response 4: Thank you for your comment. We have added the text to describe the procedure more clearly and what the purpose of all this is.

Section 2.2: “In this study, a range of measurement methods was integrated that enabled us to create high-resolution topographic data of the slot canyon. The results of UAV photogrammetry containing more texture information than the aerial scanning results can help us to preliminarily determine lithology for geomorphologic and geological research, such as revealing the genesis of the canyon. Low-altitude aerial photogrammetry using a UAV was used to obtain the point clouds to express the topographical details at the canyon top. Compared to TLS, BMLS was used to acquire point clouds to reflect the topographical characteristics of the canyon bottom and side wall due to its portability, fast and strong adaptability. In addition, Ground Control Points (GCPs) and Ground Control Planes (GCPLs) were collected to test the accuracy of the data.”

Section 2.3: Both independently acquired datasets contained local data shadows or sparse coverage that could be reduced by the fusion of the BMLS and UAV photogrammetric point clouds. Co-registration is required before fusion for aligning different point clouds. In this study, the co-registration of the point clouds in the LiDAR360® software (version 5.0) consists of manually selecting feature points for coarse registration and running an iterative closest point (ICP) algorithm for fine registration based on obtaining the initial values.  

Section 2.4: DEM, as a continuous representation of surface relief, can be generated by point clouds of a discrete digital representation [32]. In this study, the final point clouds used for generating DEM were the ground point clouds, which remained after filtering out off-ground points such as vegetation and buildings.

Point 5: The Conclusion contains a part of the text that does not belong there. It is not clear whether the results are general rules for anyone using UAV in photogrammetry or are related only to the tests performed. New findings have not been adequately highlighted and discussed. There is no comparison of results with other methods. In this way, the method cannot be valorized.

Response 5: Thank you for your comment. We have deleted the repeated text as required, and rewritten the discussion and conclusion to emphasize new findings. The comparison of results with other methods has been added in section 4.1. Two figures (Figure 13 and 14) is added to enhance the new finding and compare different methods. The results are general rules for anyone using UAV in photogrammetry. The modification is listed as follows:

(1) Section 4.1: Given the uniqueness and complexity of slot canyon geomorphological forms, such as rugged topography and the occurrence of vegetation, applying a single data acquisition method to obtain complete data in this environment is challenging. Therefore, a comprehensive data acquisition scheme is tested in this slot canyon (Figure 17). The data acquisition is divided into two parts: top data acquisition and bottom data acquisition. The canyon bottom is in a no satellite signal, short-visibility environment, the BMLS survey is used to obtain the bottom data through zigzagging along the canyon. While the data outside the upper steep terrain area uses UAV photogrammetry to obtain.

Figure 13. Comprehensive data acquisition scheme and the comparison of flight path planning methods.

To obtain the image data for 3D reconstruction, effective and safe path planning is necessary. As shown in Figure 17, the path planning based on Fixed-altitude selects the study area from a 2D satellite map without considering the impact of actual terrain and the latest data. At high altitude areas, data coverage (image) is reduced due to the reduction in relative altitude, and even the 3D reconstruction will be out due to the low overlap [44]. Path planning based on Terrain following can overcome this difficulty. It can adapt to different terrain according to DEM data, variable elevation routes can be automatically generated to keep GSD consistent and get high-quality data [45]. However, it still cannot avoid the potential safety problems of flying into the no signal area or colliding with obstacles such as the thin powerline which cannot be detected by the flight obstacle avoidance system. Meanwhile, the shooting angle is fixed in both path planning, and the corner terrain cannot be mapped finely due to the lack of data coverage. Therefore, this study developed a workflow to create the UAV fine flight path based on a large area of a rough model. Fine data acquisition was achieved through the use of twice low-altitude flights. The flight path generated was a combination of multiple waypoints, and each waypoint had its elevation and angle, allowing users to perform UAV photogrammetry faster, safer, more detailed and with higher quality.

The steps required for the fine flight of UAVs based on a rough model of the large area are as follows: The first survey of GCPs using RTK-GNSS was necessary; second, following the operational area to take flight photos and building a high-level model of the point clouds that requires little detailed knowledge of data acquisition. To ensure completeness, the accuracy (alignment step) and quality (dense point clouds step) parameters were set to Low and Lowest, respectively. Then data shadow areas were also selected in the rough model and fine flight missions were created. For fine path planning, it is necessary to mark the safe areas and dangerous areas and set the shooting parameters such as GSD and image overlap. The automatically generated KML files of the fine flight mission should be imported into the DJI pilot® or other flight path creation software to call the drone to perform a flight mission and conduct faster, safer, more detailed, and higher quality data acquisition. The workflow of fine flight path creation was shown in Figure 18.

Figure 14. The workflow of fine flight path creation.

(2) “This study has highlighted how data integration can be used to create highly accurate terrain data (centimeter-level) in the context of complex topography with local no GNSS signal area and rugged terrain, such as in slot canyon systems.

1) The integration scheme with BMLS and UAV photogrammetry applied in slot canyon systems has proven to be an effective solution for terrain reconstruction in a more accurate, complete and realistic way. In addition, multi-scale observation and planning are necessary to ensure the safety and efficiency of collecting data. Compared with path planning based on Fixed-altitude and path planning based on Terrain following, the fine flight of UAVs based on a rough model of the large area can avoid collision with obstacles such as powerlines or flying into the restricted area, allowing users to perform UAV photogrammetry faster, safer, more detailed and higher quality.

2) The registration process plays a key role in the accurate data integration of the point clouds. The accuracy of the registration by BMLS and UAV photogrammetric point clouds in this study is good with the RMSE of 0.028 m and there are no point clouds with stratification and offset. In addition, the vegetation filtering results of splitting the integrated point clouds into different slope segments used for ground point extraction are good, with all kappa coefficient values greater than 0.80.

3) The high-resolution terrain dataset achieved by data integration, including a complete color point cloud, DEM and mesh, is the starting point for generating valuable geometric parameters such as slope factor and interpreting geologic features such as the attitude and thickness of sedimentary beddings about the slot canyon system. They provide a useful supplement for revealing the morphological evolution and genesis of slot canyons.”

Point 6: Line 12-15 - This sentence is imprecise and untrue. Even before these technologies, experts studied slot canyons. Compared to previous times, today's technology makes it easier and gives better results. I guess that's what you wanted to point out.

Response 6: Thank you for your comment. We have added a precise statement to avoid ambiguity as requested.

The off-the-shelf surveying techniques, including Unmanned Aerial Vehicles (UAV) Photogrammetry and Backpack Mobile Laser Scanning (BMLS), facilitate slot canyon surveys and provide better observations.

Point 7: Line 19-20 - This is not a precise statement and cannot be expressed in this way after conducting research. You either proved something or you didn't, or you achieved better results with your method, or you didn't. This is a scientific paper and the results must be interpreted precisely.

Response 7: Thank you for your comment. We have revised these sentences to avoid ambiguity as requested.

“Data integration of BMLS and UAV photogrammetry can obtain accurate terrain datasets, with a Root Mean Squared Error (RMSE) of point cloud registration of 0.028m. Such high-resolution integration terrain datasets reduce local data shadows produced solely by individual datasets, providing a starting point for revealing the morphological evolution and genesis of slot canyons.”

Also, we have checked the full text and similar questions have been revised.

“Although such methods can provide (cm- or mm-level) accuracy and precision for measuring single points, they often provide limited, fragmented data and tend to generate data with widely varying spatial densities and uncertainties, while collecting data dense enough to generate useful terrain data would take a long time [9]. (good)”

Point 8: Line 58 - What are they? Please provide references.

Response 8: Thank you for your comment. We didn’t 't think enough, three references have been added.

“Recent advances in geospatial data collection, especially the application of active laser scanning and passive photogrammetry-based methods have greatly improved the ability for high-resolution terrain research [13,14,15].  

References:

  1. Li, L.; Wang, R.; Lin, J.; Xiao, Z.; Hui, Y. A novel approach for extraction of ripple mark parameters based on SfM. Sediment. Geol. 2019, 392, 105523.
  2. Chudá, J.; Hunčaga, M.; Tuček, J.; Mokroš, M. The handheld mobile laser scanners as a tool for accurate positioning under forest canopy. The International Archives of Photogrammetry, Remote Sensing and Spatial Information Sciences. 2020, 43, 211-8.
  3. Cucchiaro, S.; Fallu, D.J.; Zhang, H.; Walsh, K.; Van Oost, K.; Brown, A.G.; Tarolli, P. Multiplatform-SfM and TLS Data Fusion for Monitoring Agricultural Terraces in Complex Topographic and Landcover Conditions. Remote Sensing. 2020, 12, 1946.

Point 9: Line 88-89 - Another not precise and not entirely accurate statement. This statement has no place in the Introduction (if you do not provide references), but in the Conclusion with a detailed interpretation of the results.

Response 9: Thank you for your comment. We have added a precise statement and revised this paragraph to avoid ambiguity as requested.

“To obtain high-resolution terrain data of slot canyons, considering the uniqueness and complexity of geomorphological forms of slot canyons, such as variable-slope terrain, and partial vegetation cover, comprehensive data through an integration scheme is necessary to reduce local data shadows generated by a single data acquisition method.”

Point 10: Line 100-101, 103 - Is this a scientific contribution? Do you think no one else has done it before you?

Response 10: Thank you for your comment. We have reconsidered the expression and rewritten this paragraph.

“1) The fine flight of UAVs based on a rough model of the large area can avoid collision with obstacles such as powerlines or flying into a restricted area, allowing users to perform UAV photogrammetry faster, safer, more detailed and with higher quality.

2) This integration scheme greatly reduces data shadows present in the 3D point clouds produced solely by a single mapping technique, providing means for 3D mapping of extremely steep slopes and overhangs frequently present in rugged terrain. 

3) This study can provide a high-fidelity terrain dataset with sufficient spatial detail, including a complete point cloud and its derivatives such as DEM and 3D mesh, which can be used for quantitative analysis of the morphological evolution and genesis of the slot canyons.”

Point 11: Line 119 - Such type of manuscript should be written in the 3rd person.

Response 11: Thank you for your comment. We have checked the full text and similar questions have been revised. It was modified as requested:

“all the details about this workflow will be provided;

The ten times smoothing method was used for collecting the location;

the point clouds were first performed the gradient extraction and divided into;

the BMLS point clouds were applied as the reference entity and the UAV-based photogrammetric point clouds were applied as the alignment entity;

the point clouds and mesh were combined to help analyze these elevations;

the canyon development was considered at the V-shaped valley stage;

Point 12: Line 144 - Do you think you are really the first to do something like this? Also, you cite references after this claim!?

Response 12: Thank you for your comment. This is our imprecise and inaccurate statement. We have rewritten this paragraph.

Section 2.2: “In this study, a range of measurement methods was integrated that enabled us to create high-resolution topographic data of the slot canyon. The results of UAV photogrammetry containing more texture information than the aerial scanning results can help us to preliminarily determine lithology for geomorphologic and geological research, such as revealing the genesis of the canyon. Low-altitude aerial photogrammetry using a UAV was used to obtain the point clouds to express the topographical details at the canyon top. Compared to TLS, BMLS was used to acquire point clouds to reflect the topographical characteristics of the canyon bottom and side wall due to its portability, fast and strong adaptability. In addition, Ground Control Points (GCPs) and Ground Control Planes (GCPLs) were collected to test the accuracy of the data.”

Point 13: Line 362 - Which? Please provide references.

Response 13: Thank you for your comment. This is our negligence. We have added three references. The modification is listed as follows:

“Many published papers related with filtering algorithms adopted a manually revised method to generate the reference data to analyze the accuracy of algorithms. Therefore, this study adopted the revised classification results as reference data [34, 38-39].”

“References:

34     Zhao, X.; Guo, Q.; Su, Y.; Xue, B. Improved progressive TIN densification filtering algorithm for airborne LiDAR data in forested areas. ISPRS J. Photogramm. 2016, 117, 79-91.

  1. Guan, H.; Li, J.; Yu, Y.; Zhong, L.; Ji, Z. DEM generation from lidar data in wooded mountain areas by cross-section-plane analysis. Int. J. Remote Sens. 2014, 35, 927-48.
  2. Mongus, D.; Žalik, B. Parameter-free ground filtering of LiDAR data for automatic DTM generation. ISPRS-J. Photogramm. 2012, 67, 1-12.

Point 14: Line 584-589 - This text has no place in the Conclusion. The facts from the previous text are repeated here.

Response 14: Thank you for your comment. We have deleted the repeated text and revised this section.

This study has highlighted how data integration can be used to create highly accurate terrain data (centimeter-level) in the context of complex topography with local no GNSS signal area and rugged terrain, such as in slot canyon systems.

1) The integration scheme with BMLS and UAV photogrammetry applied in slot canyon systems has proven to be an effective solution for terrain reconstruction in a more accurate, complete and realistic way. In addition, multi-scale observation and planning are necessary to ensure the safety and efficiency of collecting data. Compared with path planning based on Fixed-altitude and path planning based on Terrain following, the fine flight of UAVs based on a rough model of the large area can avoid collision with obstacles such as powerlines or flying into the restricted area, allowing users to perform UAV photogrammetry faster, safer, more detailed and higher quality.

2) The registration process plays a key role in the accurate data integration of the point clouds. The accuracy of the registration by BMLS and UAV photogrammetric point clouds in this study is good with the RMSE of 0.028 m and there are no point clouds with stratification and offset. In addition, the vegetation filtering results of splitting the integrated point clouds into different slope segments used for ground point extraction are good, with all kappa coefficient values greater than 0.80.

3) The high-resolution terrain dataset achieved by data integration, including a complete color point cloud, DEM and mesh, is the starting point for generating valuable geometric parameters such as slope factor and interpreting geologic features such as the attitude and thickness of sedimentary beddings about the slot canyon system. They provide a useful supplement for revealing the morphological evolution and genesis of slot canyons.”

Round 2

Reviewer 1 Report (Previous Reviewer 3)

Dear authors,

Thank you for addressing the comments. The last but not the least suggestion is to carefully check the English fluency of the text.

Success!

Author Response

Response to Reviewer 1 Comments

Point 1: Dear authors,

Thank you for addressing the comments. The last but not the least suggestion is to carefully check the English fluency of the text.

Success!

Response 1: We gratefully appreciate your affirmation of our modification and give us more valuable suggestions. Based on your comment, we have carefully checked the manuscript sentence by sentence, and made some language improvements, including the sentences are not fluent and the words are inappropriate. (The detailed modifications are demonstrated in the revised manuscript. Revisions are marked up using the “Track Changes”).

This manuscript is a resubmission of an earlier submission. The following is a list of the peer review reports and author responses from that submission.

Round 1

Reviewer 1 Report

Dear Authors,

Thank you very much for your wonderful article, which I think is a superb application example. The comparison of the two surveying methods is very well presented and results in a perfect and solid case study.
I really like the approach of combining aerial and ground surveying to get a high resolution digital surface of such a challenging environment and I also find the presentation of the results very well done.

The article is written in a good and easily understandable English. The structure is well thought out and the illustrations support the statements of the text.

All in all, it is a great article that I am very happy to suggest for publication.

Reviewer 2 Report

it would be useful to integrate the text with more images. In particular, it would be useful to be able to see images that describe the results of the point clouds generated by the BMLS by inserting in the text a repertoire of images that make one appreciate the different qualities of the data acquired and then processed. Images could also help to clarify the description of the different steps of the methodology allowing to appreciate the quality of the product.

Reviewer 3 Report

The manuscript presents a strategy to improve 3d information generation from Slot Canyon considering their spatial structure. Different technologies are combined for the purpose. My main concern here is the motivation and the scientific aspect. However, some comments have been added to the manuscript for your reference.

Reviewer 4 Report

Accurate terrain models are critical for many applications. This study focuses on fusing BMLS and UAV photogrammetry points for the terrain reconstruction of slot canyon which has some challenges in deed.  Maybe these two followed improvements should be done for more distinguish:

1. Please add the accuracy accuracy for filtering such as Type I, Type II error.

2. The co-registration of BMLS and UAV points is very imprtant. Please add more details about the co-registration between two different data sources including side wall and top view.